# Modeling the impact of public response on the COVID-19 pandemic in Ontario

**Brydon Eastman** *, **Cameron Meaney**, **Michelle Przedborski, Mohammad Kohandel**

Department of Applied Mathematics, University of Waterloo, Waterloo, Ontario, Canada

* b2eastma@uwaterloo.ca

**Data Availability Statement:** All relevant data files are available from the GitHub database https://github.com/ishaberry/Covid19Canada.

**Funding:** The authors received no specific funding for this work.

## Abstract

The outbreak of SARS-CoV-2 is thought to have originated in Wuhan, China in late 2019 and has since spread quickly around the world. To date, the virus has infected tens of millions of people worldwide, compelling governments to implement strict policies to counteract community spread. Federal, provincial, and municipal governments have employed various public health policies, including social distancing, mandatory mask wearing, and the closure of schools and businesses. However, the implementation of these policies can be difficult and costly, making it imperative that both policy makers and the citizenry understand their potential benefits and the risks of non-compliance. In this work, a mathematical model is developed to study the impact of social behaviour on the course of the pandemic in the province of Ontario. The approach is based upon a standard SEIRD model with a variable transmission rate that depends on the behaviour of the population. The model parameters, which characterize the disease dynamics, are estimated from Ontario COVID-19 epidemiological data using machine learning techniques. A key result of the model, following from the variable transmission rate, is the prediction of the occurrence of a second wave using the most current infection data and disease-specific traits. The qualitative behaviour of different future transmission-reduction strategies is examined, and the time-varying reproduction number is analyzed using existing epidemiological data and future projections. Importantly, the effective reproduction number, and thus the course of the pandemic, is found to be sensitive to the adherence to public health policies, illustrating the need for vigilance as the economy continues to reopen.

## Introduction

On January 11, 2020, China reported the first death due to a novel strain of coronavirus, now officially named SARS-CoV-2, to the World Health Organization (WHO) [1]. To date, the origin of the virus remains uncertain; however, it is thought to have originated in a wet market in Wuhan, China via inter-species transmission to humans. By January 20, 2020, the number of infections in China had sharply risen to 278, including six deaths, and several cases had also been confirmed in Japan, South Korea and Thailand [1]. The virus quickly spread to North America through intercontinental travel: the CDC reported the first case in the State of

**Competing interests:** The authors have declared that no competing interests exist.

Washington, United States on January 21, 2020 [2], and four days later Canada confirmed its first case in Toronto, Ontario [3].

As of August 2020, epidemiological data curated by Johns Hopkins Center for Systems Science and Engineering indicates that over 19 million people have been infected globally by SARS-CoV-2 and over 700,000 people have died as a result of complications from COVID-19, the disease caused by the virus. The virus is now affecting people in 215 countries and territories around the globe. To attenuate the spread of the virus, several different strategies for reducing transmission have been implemented in countries around the world. These strategies have ranged from quarantining infected individuals and those returning from international travel, to orders to avoid public parks and facilities, to prohibiting gatherings of more than two people (with exceptions for family members), to mandating the use of masks in public spaces. In addition, in some countries, all day cares, educational facilities, and non-essential businesses were ordered temporarily closed by government officials, and citizens were ordered to stay home, except to go shopping for essential supplies, such as groceries and prescriptions. In South Korea, for instance, a recent report indicated that social distancing protocols were responsible for reducing the effective reproduction number in several cities (below 1 in some cases) by estimating transmission-reducing effects via traffic or metro data [4].

Data obtained by tracking mobile phone locations [5, 6] indicates that a fair proportion of people are abiding by government implemented virus mitigation policies in North America. In particular, data collected for several weeks between March 2020 and mid-April 2020, when government measures were most strict, indicated that mobility toward the workplace decreased between 24% and 52% in different parts of North America while mobility toward retail and recreation decreased between 29% and 70% [5, 6]. These data raise the questions of how these policies have modulated the timeline of SARS-CoV-2 outbreaks, how abiding by guidelines have mitigated the overload to the medical system, and how sensitive these effects are to the proportion of the population abiding by the policies.

To address these and other questions, we developed a mathematical model to study the effects of transmission-reduction guidelines on the spread of SARS-CoV-2, with an emphasis on the policies implemented in Ontario, Canada as a focused example. First, using machine learning techniques, we fit the temporal evolution of infection counts, total recoveries, and total deaths in Ontario, obtained from Ref. [7], to an SEIRD infection model. Then we use the fitted parameters as the basis for an analysis of the model behaviour, including predicting the occurrence of a second wave, the qualitative behaviour of candidate future strategies, and a prediction of the basic and time-varying reproduction number corresponding to past data and future projections.

The manuscript is organized as follows. In the Materials and methods section, we describe the model used in this work and give a brief discussion of the techniques that were used to estimate the SEIRD model parameters to fit the temporal SARS-CoV-2 data in Ontario. In the Results section, we present our results and give a discussion of the significant model predictions. Specifically, we investigate the conditions that lead to the emergence of a second wave of infections, as well as how the disease behaves under different future transmission-reduction strategies, and how these different strategies affect the reproduction number of the disease. Finally, in the Conclusion we discuss the results and provide some concluding remarks and future research directions.

## Materials and methods

### Mathematical model

The most common method for the mathematical modelling of infectious disease spread involves dividing the population into distinct compartments and quantifying the flow between

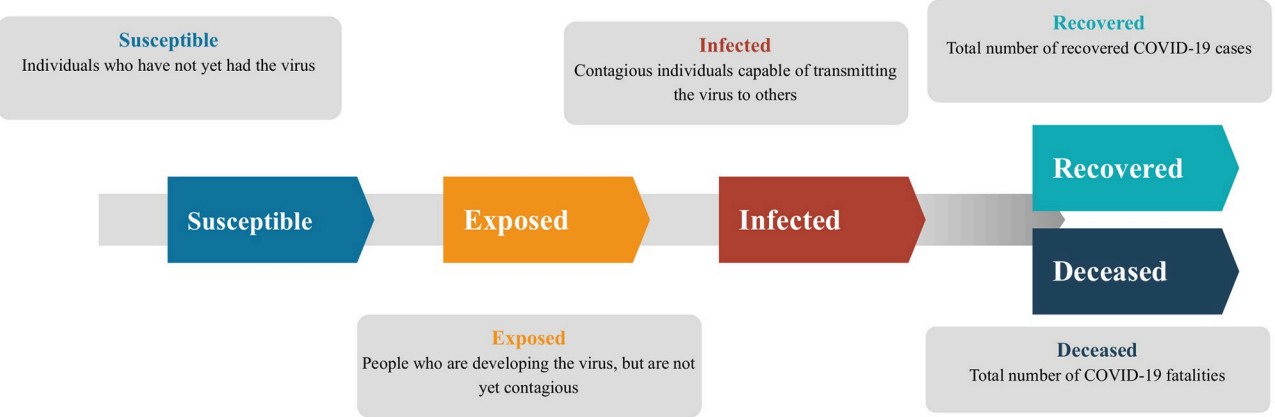

**Fig 1. Schematic of the standard SEIRD epidemiological model.** The total population is divided into five compartments: susceptible, exposed, infected, recovered, and deceased individuals.

the compartments. One example of this is the standard SEIRD model that is illustrated in Fig 1. In this model, it is assumed that all individuals comprising the total population fall into one of five categories: susceptible (S), exposed (E), infected (I), recovered (R), or deceased (D). The "susceptible" compartment refers to individuals who have never been infected by the disease and therefore lack antibodies to defend against it. The "exposed" compartment consists of those who have caught the disease from an infected individual, but have not yet become contagious. The "infected" individuals are those who have the disease and are now contagious. The "recovered" compartment refers to individuals who have gone through the whole life-cycle of the disease and survived. For viral infections, it is often assumed that recovered individuals have a prepared immune response and will not become infected again. Finally, the "deceased" compartment represents the cumulative number of fatalities resulting from the disease. In this model, some individuals may never leave the susceptible compartment during the outbreak of the disease, which represents those who do not contract the disease at all. However, once an individual enters the exposed compartment, they will irreversibly progress to the infected compartment and subsequently to either the recovered or deceased compartment. We assume that anyone who has been exposed to enough pathogen as to become contagious will eventually become infectious and progress through the entire life cycle of the disease. Individuals who are exposed to the virus but not enough to become themselves contagious are still captured by the model, as we assume they have not built up enough antibodies so as to trigger an immune response. As a result, those such individuals remain in the susceptible compartment.

In this work, we use a generalized SEIRD model to quantify the spread of SARS-CoV-2 among the population of Ontario, Canada. The model consists of five compartments whose time evolution is described by Eq (1): $S(t)$ is the population of individuals that are susceptible to the virus, $E(t)$ is the population of individuals who have been exposed to the virus but are not yet contagious, $I(t)$ is the population of individuals that are infected by the virus, $R(t)$ is the population of individuals who have recovered and thus have immunity, and $D(t)$ is the number of fatalities from the virus. We further assume that the $S$, $E$, and $I$ compartments are stratified into "distancer" and "mixer" sub-compartments. The "distancer" sub-compartment, denoted with a subscript $D$, refers to those who are adhering to public health guidelines. The "mixer" sub-compartment, denoted with a subscript $M$, refers to those who are not adhering to public health policies. Mathematically, this means that $S = S_D + S_M$, $E = E_D + E_M$, and $I = I_D + I_M$. The $R$ and $D$ compartments are not explicitly subdivided into distancer and mixer categories since

individuals do not leave these compartments once they enter them. While the stratification devised here can loosely be thought of as merely distinguishing between those who are and are not practicing social distancing guidelines, more strictly speaking, it delineates across all public health transmission-reduction guidelines. These include the wearing of masks in public spaces, adhering to a strict social-bubble, frequent hand washing, and other public health measures.

$$
\begin{aligned}
\frac{dS}{dt} &= \underbrace{-\beta_{DD}\,S_D\,I_D}_{\text{D-D Exposure}} - \underbrace{\beta_{DM}\,S_D\,I_M}_{\text{D-M Exposure}} - \underbrace{\beta_{MD}\,S_M\,I_D}_{\text{M-D Exposure}} - \underbrace{\beta_{MM}\,S_M\,I_M}_{\text{M-M Exposure}} \\
\frac{dE}{dt} &= \underbrace{\beta_{DD}\,S_D\,I_D}_{\text{D-D Exposure}} + \underbrace{\beta_{DM}\,S_D\,I_M}_{\text{D-M Exposure}} + \underbrace{\beta_{MD}\,S_M\,I_D}_{\text{M-D Exposure}} + \underbrace{\beta_{MM}\,S_M\,I_M}_{\text{M-M Exposure}} \\
\frac{dI}{dt} &= \underbrace{\gamma_D\,E_D + \gamma_M\,E_M}_{\text{Disease Progression}} - \underbrace{(\alpha_D + \delta_D)\,I_D - (\alpha_M + \delta_M)\,I_M}_{\text{Recovery/Death}} \\
\frac{dR}{dt} &= \underbrace{\alpha_D\,I_D + \alpha_M\,I_M}_{\text{Recovery}} \\
\frac{dD}{dt} &= \underbrace{\delta_D\,I_D + \delta_M\,I_M}_{\text{Death}}
\end{aligned}
\tag{1}
$$

To simplify the model, we make the assumption that the proportion of people that are adhering to public health guidelines is given by the time-dependent function $\theta(t)$. Mathematically, we impose the following conditions on the stratified $S$, $E$, and $I$ sub-compartments:

$$
\begin{aligned}
S_D &= \theta(t)\,S \\
S_M &= (1 - \theta(t))\,S \\
E_D &= \theta(t)\,E \\
E_M &= (1 - \theta(t))\,E \\
I_D &= \theta(t)\,I \\
I_M &= (1 - \theta(t))\,I
\end{aligned}
\tag{2}
$$

Making this assumption, Eq (1) is reduced to the following form, which we refer to as the Distancing-SEIRD model:

$$
\begin{aligned}
\frac{dS}{dt} &= \underbrace{-\beta(\theta(t))\,S\,I}_{\text{Contact with Infection}} \\
\frac{dE}{dt} &= \underbrace{\beta(\theta(t))\,S\,I}_{\text{Contact with Infection}} - \underbrace{(\gamma_D\,\theta(t) + \gamma_M\,(1-\theta(t)))\,E}_{\text{Disease Progression}} \\
\frac{dI}{dt} &= \underbrace{(\gamma_D\,\theta(t) + \gamma_M\,(1-\theta(t)))\,E}_{\text{Disease Progression}} - \underbrace{((\alpha_D + \delta_D)\,\theta(t) + (\alpha_M + \delta_M)\,(1-\theta(t)))\,I}_{\text{Recovery/Death}} \\
\frac{dR}{dt} &= \underbrace{(\alpha_D\,\theta(t) + \alpha_M\,(1-\theta(t)))\,I}_{\text{Recovery}} \\
\frac{dD}{dt} &= \underbrace{(\delta_D\,\theta(t) + \delta_M\,(1-\theta(t)))\,I}_{\text{Death}}
\end{aligned}
\tag{3}
$$

where

$$\beta(\theta(t)) = \beta_{DD}\,\theta(t)^2 + (\beta_{DM} + \beta_{MD})\,\theta(t)\,(1 - \theta(t)) + \beta_{MM}\,(1 - \theta(t))^2 \qquad (4)$$

is a time-dependent transmission rate that depends on the proportion of people who are adhering to public health guidelines.

The parameters $\beta_{DD}$, $\beta_{DM}$, $\beta_{MD}$, and $\beta_{MM}$ have been introduced in Eqs (1)–(4) to quantify the rate of virus transmission from infected to susceptible individuals, while taking into account the transmission-reduction practices. Specifically, $\beta_{DD}$ describes the rate of transmission from infected distancers to susceptible distancers, $\beta_{DM}$ from infected distancers to susceptible mixers, $\beta_{MD}$ from infected mixers to susceptible distancers, and $\beta_{MM}$ from infected mixers to susceptible mixers. It is expected that individuals who adhere to public health guidelines are less likely to come into contact with others and transmit the virus. Thus, we assume that $\beta_{DD} < \beta_{DM} = \beta_{MD} < \beta_{MM}$. These assumptions imply that distancers are least likely to pass the disease to one another and that mixers are most likely to pass the disease to one another. Furthermore, by assigning $\beta_{DM} = \beta_{MD}$, we make the simplifying assumption that transmission between an infected distancer and a susceptible mixer is the same as transmission between an infected mixer and a susceptible distancer. Moreover, we assume that this cross-compartment transmission rate falls between the two inter-compartment transmission rates. Similarly, while the pairs of parameters $\gamma_M$ and $\gamma_D$, $\alpha_M$ and $\alpha_D$, and $\delta_M$ and $\delta_D$ are theoretically distinct, we assume that each pair is the same across sub-compartments, i.e. $\gamma_M = \gamma_D \equiv \gamma$, $\alpha_M = \alpha_D \equiv \alpha$, and $\delta_M = \delta_D \equiv \delta$. Effectively, we make the assumption that transmission-reduction strategies primarily impact the transmission rate $\beta$, but not the latent period $\gamma^{-1}$ or the mean infectious length $(\alpha + \delta)^{-1}$, which are both instead dictated entirely by the disease itself.

The interpretation of each term in the Distancing-SEIRD model is described in Eq (3). Given the assumptions used in this work, the Distancing-SEIRD model depends upon four distinct kinetic parameters. The time-dependent parameter $\beta$ controls the rate of virus transmission from infected to susceptible individuals. It is determined by the probability of disease transmission as well as the chance of contact, thus it indirectly incorporates the basic reproduction number, $R_0$, and it depends explicitly on the proportion of individuals who are adhering to public health guidelines, as indicated in Eq (4). The mean latent period, given by $\gamma^{-1}$, is the average length of time between exposure to the virus and the point at which an individual becomes contagious. The rate at which infectious individuals are removed from the disease, either via recovery or death, is given by $(\alpha + \delta)$. Thus the value of $(\alpha + \delta)^{-1}$ is the mean length of time an infected individual is contagious before they either recover from or die from the disease. The five compartments are subjected to the constraint $S + E + I + R + D = N$, where $N$ is the total population. Importantly, the state equations are considered to represent direct counts of the population size in this work, not the proportions of the population in each compartment.

In contrast to other approaches that have been proposed to model the effects of social distancing and self isolation on the spread of SARS-CoV-2, see for example Ref. [8], the model developed in this work captures two infection pathways that weakly interact with each other. Distinctly, the distancing compartment in the model proposed here does not represent a strict quarantine, since we assume that distancing individuals are still engaging in public life while following public health protocols. Furthermore, the Distancing-SEIRD model developed here is a simplistic, first generation model that does not include age, population density, geographical distribution, environmental conditions, loss of immunity, or multiple strains of the virus. Importantly, recent genomic data indicates that a mutated form of the virus, carrying the

Spike protein amino acid change D614G, comprises the majority of cases worldwide and has a higher infectivity than the ancestral strain of SARS-CoV-2 [9].

As powerful as mathematical models are at predicting the spread of infectious diseases, all modelling is subject to simplifying assumptions to remain tractable. In addition to the assumptions stated above, the model developed here does not consider how the burden on the healthcare system directly affects the case fatality rate. In reality, it would be expected that if the healthcare system is overburdened by a large number of simultaneous infections, the case fatality rate should increase. This could result from health care practitioners being unable to provide care for all critically-ill patients due to limitations in hospital capacity and staff or medical equipment and treatments. Consequently, even with effective triage procedures, it is anticipated that more fatalities would result from COVID-19 disease in an overburdened healthcare system compared to one with extra resources, even when the total case count is constant over a period of time.

Further, in this work we ignore the role of random events, and assume that the population is well-mixed and that the spread of the virus throughout the province behaves as a single outbreak. In reality, random events can influence the spread of the virus in profound ways, populations are quite heterogeneous and not well mixed, and, especially in the early stages of an outbreak, the spread of the virus is often characterized by multiple smaller outbreaks happening out of sync with one another. As an epidemic outbreak progresses, the validity of these assumptions changes. For instance, in the early days of a viral outbreak in a particular province, we might observe smaller, discrete outbreaks in multiple cities evolving without any influence from one another; however, as those outbreaks grow in size, they eventually overlap and further behave as a single outbreak.

## Parameter estimates from fitting to epidemiological data

The Distancing-SEIRD model, Eqs (3) and (4), requires calibration with meaningful values of several mathematical parameters. The parameter values can be estimated by fitting to epidemiological data, then the calibrated model can be used to forecast the spread of SARS-CoV-2 in the population. The epidemiological data we use for calibrating the model is collated by the COVID-19 Canada Open Data Working Group [7]. Effectively, this data set has the form $I_t$, $R_t$, $D_t$ for $t \in \{0 = t_1, t_2, \ldots, t_{n-1}, t_n = 157\}$, where $I_{t_i}$ represents the active known infections at time $t_i$, $R_{t_i}$ the known cumulative recoveries by time $t_i$, and $D_{t_i}$ the cumulative known fatalities at time $t_i$. The time points of the data series correspond to each day between January 25, 2020 ($t_1 = 0$) and June 30, 2020 ($t_n = 157$). To calibrate the model to this data, we introduce an additional non-model parameter $x_v$, which can be interpreted as the visible infective proportion. This value represents the proportion of active infected cases that have a confirmed diagnosis. In particular, $1 - x_v$ represents the proportion of people who are actively infected but who have not received a positive COVID-19 diagnosis. This number could be due to asymptomatic or mildly symptomatic individuals who never approached public health officials for diagnosis or from results that were false negatives due to errors in the testing. In reality, the number of daily tests performed is time-dependent, so we would expect $x_v$ to also exhibit time dependence; however, for simplicity, it is fit here to a constant value.

There are many possible functional forms for $\theta(t)$, the function which represents the population's adherence to public health guidelines. Here, we explain our rationale for our chosen form. The province of Ontario declared a state of emergency on March 17, 2020. As a result, we assume that $\theta(t) = 0$ for all time points before March 17, 2020; that is, $\theta(t) = 0$ for $t \leq 52$ (since the first confirmed infection in Ontario occurred on January 25, 2020 [3]. For the remaining 105 days between March 17, 2020 and June 30, 2020 we expect $\theta(t)$ to vary

continuously. This represents the varying response of the citizenry to public health guidelines. This variation in public response could be due to new public health guidelines, the re-opening of various aspects of the economy, or even individual variation for personal reasons. To describe $\theta(t)$ for these remaining 105 days we subdivide that time into 7 equal intervals of 15 days each and make the simplifying assumption that $\theta(t)$ is a linear spline interpolating the points $(\tau_i, \theta_i)$ for $i \in \{0, \ldots, 7\}$ where $\tau_0 = 52$ and $\tau_7 = 157$. Hence, $\theta(t)$ is a continuous, piece-wise-linear function for $t_1 \leq t \leq t_n$, described by the following form:

$$\theta(t) \equiv \begin{cases} \theta_0 & \text{if } t_1 \leq t \leq \tau_0 \\ \theta_k + (\theta_{k+1} - \theta_k) \dfrac{t - \tau_k}{\tau_{k+1} - \tau_k} & \text{if } \tau_k < t \leq \tau_{k+1} \text{ for } k \in \{0, \ldots, 7\} \end{cases} \tag{5}$$

To fit the model to the epidemiological data, we define a loss function that is minimized via differential evolution [10]. For a given parameter set $\xi = (\gamma, \alpha, \delta, \theta_1, \ldots, \theta_7, \beta_{DD}, \beta_{DM}, \beta_{MM}, x_v)$, the loss function $\chi(\xi)$ is defined as the weighted average of the following quantities: the mean squared error (MSE) between visible infectious cases in the model and in the data, the MSE between recovered cases in the model and the data, and the MSE between fatalities in the model and the data. Explicitly, $\chi(\xi)$ has the form

$$\chi(\xi) = \frac{1}{n}(W_I + W_R + W_D)^{-1} \left( W_I \sum_{i=0}^{n} (x_v I(t_i) - I_{t_i})^2 \right.$$
$$\left. + W_R \sum_{i=0}^{n} (R(t_i) - R_{t_i})^2 + W_D \sum_{i=0}^{n} (D(t_i) - D_{t_i})^2 \right), \tag{6}$$

where $I(t)$, $R(t)$, $D(t)$ are trajectories of Eq (3) parameterized by $\xi$. The weights $W_I$, $W_R$, and $W_D$ were chosen to be $W_I = 10$, $W_R = 1$, $W_D = 1$, giving primary importance to deviations from the reported infection counts in the fitting process.

In Fig 2, we present a plot of the epidemiological data along with the fit trajectories of the Distancing-SEIRD model, Eqs (3) and (4), acquired by this fitting procedure. The figure depicts a noticeable difference between visible and total infections, where the visible infections is simply equal to the number of total infections multiplied by the proportion of visible cases, $x_v$. Importantly, in the fitting process, the epidemiological time-series $I_t$ is considered to be strictly the visible infections. The full set of parameter values, $\xi^*$, resulting from this fitting procedure is presented in Table 1. The corresponding predicted public-response function is illustrated in Fig 3 and depicts two prominent peaks. A local sensitivity analysis of the model, conducted around the fit parameter values, is presented in Appendix 1.

As mentioned in the Mathematical Model section, some of the mathematical parameters have a direct biological interpretation. For example, the reciprocal of the parameter $\gamma$ is an estimate of the latency period of the disease. Based on our fitted estimate for $\gamma$, this corresponds to roughly four days, in agreement with other mathematical models [11] and slightly below a report that puts the 95% confidence interval for the latency period at 4.5-5.8 days [12]. Similarly, the procedure predicted values for $\alpha$ and $\delta$ that correspond to an infectious period (the mean time from infection to either recovery or death) of roughly 18 days. In Ontario, a non-hospitalized case is marked as recovered exactly 14 days past symptom onset. As a result, the infectious period reflected in the epidemiological data is expected to be greater than 14 days, especially since more serious hospitalized cases can take longer than 14 days to resolve in some individuals. For instance, the World Health Organization reports that the time from onset to recovery is around two weeks for mild cases but is 3-6 weeks for patients with severe or critical

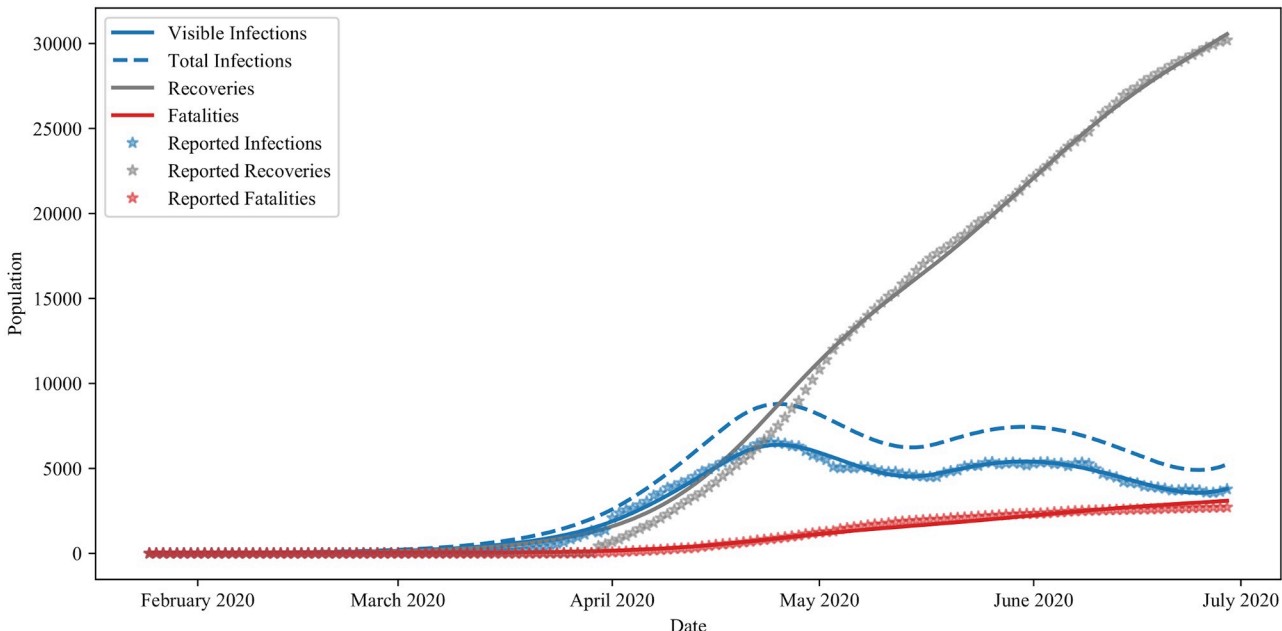

**Fig 2. Ontario model fitting.** A comparison of Ontario time-series data with the predictions of the Distancing-SEIRD model obtained using the fit parameters in Table 1.

disease [13]. Furthermore, the fitting procedure also determined a value for $\theta_1$ on the order of $10^{-5}$ suggesting that there was not a large reduction of transmission during the first 15 days after the province declared a state of emergency. There are a variety of explanations for this beyond just accounting for the latent period of the disease. For example, while March 17, 2020 was the date Ontario entered state of emergency, non-essential businesses were closed in waves on March 24, 2020 and April 4, 2020. The procedure predicted an $x_v$ value of roughly 0.73, corresponding to 27% of cases not being reported. Finally, the predicted values of $\beta_{DD}$ and $\beta_{DM}$ differ by roughly, 5%, whereas there is a predicted nearly nine-fold reduction in transmission between $\beta_{DM}$ and $\beta_{MM}$. As such, the calibrated model appears to suggest that as long as an individual is mindful of transmission-reduction guidelines, they are protected against transmission even from those who eschew such public health mandates.

## Prediction of a second wave of infections

One of the key goals in planning future disease-control measures is to prevent the occurrence of a second wave of infections. Moreover, once appropriate public health measures are established, prevention of a second wave requires public adherence to these policies. Thus far, the public response function $\theta(t)$, presented in Eq (5), is only defined for $t \in [t_1, t_n]$. However, to mathematically predict a second wave, it is necessary to consider $\theta(t)$ for $t > t_n$.

More generally, given a prescribed function of $\theta(t)$ for $t \leq t_c$, we wish to predict the effects of $\theta(t)$ for $t > t_c$ on the control of an additional wave of infection. While the form of $\theta(t)$ for $t > t_c$ (hereafter denoted $\theta(t > t_c)$) is unlikely to be constant (as human behaviour is unlikely to be constant), to begin, we assume that future $\theta(t)$ takes a constant value, denoted $\theta(t > t_c) = \theta_f$. Then, to impose the absence of a second wave, we search for a condition on $\theta_f$ that ensures that the number of active cases do not increase for $t > t_c$. That is, we wish to show to find a condition on $\theta_f$ such that $I'(t > t_c) \leq 0$. In Appendix 2 we demonstrate that such a condition is only possible if $I'(t_c) \leq 0$. Moreover, we show that the condition separates the interval [0, 1] into

**Table 1. Parameter values for the Distancing-SEIRD model.** Parameter values for Eqs (3) and (4), and the public-response function, Eq (5), obtained by fitting to COVID-19 epidemiological data from Ontario between January 25, 2020 and June 30, 2020 inclusive.

| Chosen Parameter | Value | Unit |
|---|---|---|
| $N$ | 14711827 | person |
| $E_0$ | 20 | person |
| $I_0$ | 1 | person |
| $\theta_0$ | 0 | — |
| $\tau_0$ | 52 | day |
| $\tau_1$ | 67 | day |
| $\tau_2$ | 82 | day |
| $\tau_3$ | 97 | day |
| $\tau_4$ | 112 | day |
| $\tau_5$ | 127 | day |
| $\tau_6$ | 142 | day |
| $\tau_7$ | 157 | day |
| Fit Parameter | Value | Unit |
| $\gamma$ | $2.50064531 \times 10^{-1}$ | $(\text{day}^{-1})$ |
| $\alpha$ | $5.00023339 \times 10^{-2}$ | $(\text{day}^{-1})$ |
| $\delta$ | $5.06556146 \times 10^{-3}$ | $(\text{day}^{-1})$ |
| $\beta_{DD}$ | $1.37695894 \times 10^{-9}$ | $(\text{person}^{-1})(\text{day}^{-1})$ |
| $\beta_{DM}$ | $1.38384205 \times 10^{-9}$ | $(\text{person}^{-1})(\text{day}^{-1})$ |
| $\beta_{MM}$ | $1.22423227 \times 10^{-8}$ | $(\text{person}^{-1})(\text{day}^{-1})$ |
| $x_v$ | $7.26417750 \times 10^{-1}$ | — |
| $\theta_1$ | $7.52722826 \times 10^{-5}$ | — |
| $\theta_2$ | $2.38473832 \times 10^{-1}$ | — |
| $\theta_3$ | $9.98963875 \times 10^{-1}$ | — |
| $\theta_4$ | $3.33439201 \times 10^{-1}$ | — |
| $\theta_5$ | $5.95414821 \times 10^{-1}$ | — |
| $\theta_6$ | $9.15801476 \times 10^{-1}$ | — |
| $\theta_7$ | $1.02299056 \times 10^{-1}$ | — |

two sub intervals $[\theta_{\text{critical}})$ and $[\theta_{\text{critical}}, 1]$ such that if $\theta_f \in [0, \theta_{\text{critical}})$, then another wave of infection is inevitable whereas if $\theta \in [\theta_{\text{critical}}]$, then another wave of infections is avoided.

Further details of this process and the resulting algorithm for determining the conditions under which such a $\theta_{\text{critical}}$ exists, and how to arbitrarily approximate this $\theta_{\text{critical}}$ value, can be found in Appendix 2.

## Results

### Occurrence of a second wave depends critically on public response

For the fitted parameter values, see Table 1, that correspond to Ontario's epidemiological data as of $t_c = \tau_6$ (June 15, 2020), the critical value of $\theta$ is given as $\theta_{\text{critical}} \approx 0.5326$. Importantly, this analysis is not unique to the case of Ontario—a similar number could be obtained for any population, given estimates of the model parameters for that population and recent infection data. Here we have chosen to take the infection as of June 15, 2020 and project forward as, at that time, the infection count in the model was trending downward. Importantly, as can be seen in Appendix 2, if $I'(t_c) > 0$, then another wave of infection is inevitable.

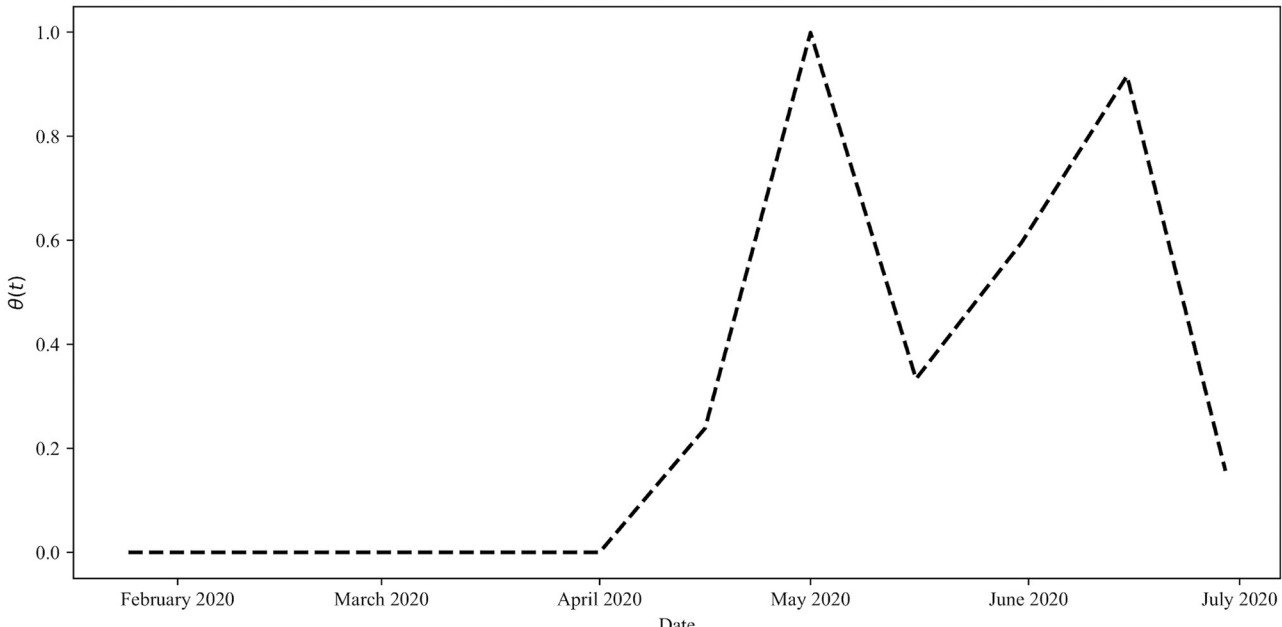

**Fig 3. Public-response function fitting.** The public-response function $\theta(t)$ between January 25, 2020 and June 30, 2020. The function is a continuous, piecewise-linear function with intervals of 15 days.

To demonstrate the sharp sensitivity of this condition, we plot visible infection trajectories of the Distancing-SERID model, Eqs (3) and (4), in Fig 4 for three separate values of $\theta_f$. All three trajectories use the same parameter set from Table 1 and the form of $\theta(t)$ from Eq (5) with the addition that $\theta(t) = \theta_f$ for $t > t_c$. However, in the first trajectory, we consider

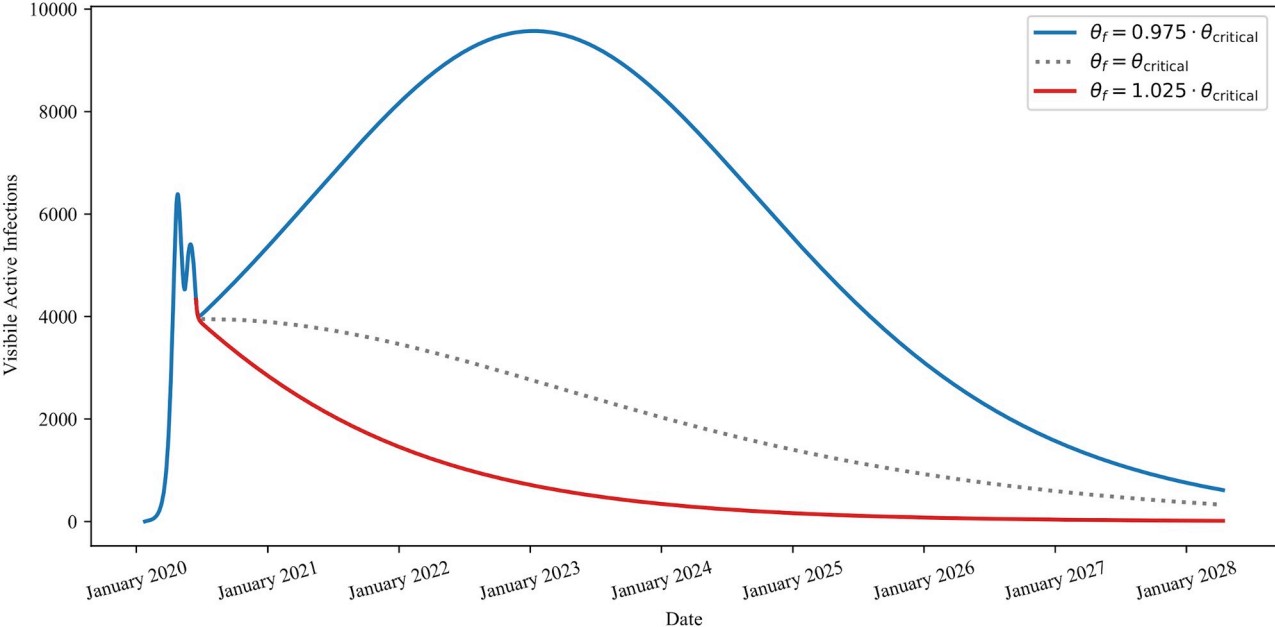

**Fig 4. Behaviour of infections for $\theta$ at, above, and below $\theta_{\text{critical}}$.** Note how all three trajectories initially decrease for $t > t_c = \tau_6$ (June 15, 2020). In the case where $\theta_f > \theta_{\text{critical}}$ the red line decreases for all $t > t_c$ whereas in the case where $\theta_f < \theta_{\text{critical}}$ the blue line initially decreases before eventually increasing again, seeding a second wave of infection. Finally, in the case where $\theta = \theta_{\text{critical}}$ the dashed line initially decreases, levels out, and then continues decreasing to zero.

$\theta_f = 0.975\ \theta_{\text{critical}}$, in the second trajectory we consider $\theta_f = \theta_{\text{critical}}$, and in the third trajectory we consider $\theta_f = 0.1.025\ \theta_{\text{critical}}$.

The scenarios presented in Fig 4 demonstrate the sharp, and non-symmetric, response to perturbations in $\theta_f$ about $\theta_{\text{critical}}$. That is, the number of additional active cases observed after a 2.5% perturbation below $\theta_{\text{critical}}$ far outweigh the number of cases avoided after a 2.5% perturbation above $\theta_{\text{critical}}$. This fragility in the development of a second wave of infections demonstrates the need to err on the side of caution while in social and public settings. The long timescale of the simulations in Fig 4 is a result of how close the value of $\theta_f$ is to the value of $\theta_{\text{critical}}$. For instance, if $\theta_f$ was instead taken to be 20% larger than $\theta_{\text{critical}}$ (resulting in $\theta_f \approx 0.64$) the downward slope of the infection trajectory is much sharper, and the model predicts less than 10 active cases in the province by June 2021.

## Time-varying containment strategies can "flatten the curve"

An effective strategy for combating the spread of the virus is one that causes the current infection number to decrease over time and the cumulative deaths and recoveries to plateau. By solving the Distancing-SEIRD model, Eqs (3) and (4), for future $\theta(t)$ schedules, where $t > t_n = 157$, we can make qualitative, short-term predictions about the nature of the pandemic beyond the current epidemiological data. In Fig 5, we present the temporal evolution of the infections and deaths for six possible future $\theta(t)$ schedules. The schedules were chosen to represent prospective plans that have been considered for the coming months. As discussed above, $\theta_{\text{critical}}$ can be found at any given point in time to determine whether the infection numbers will increase or decrease beyond that point. For $t = t_n = 157$ days and the parameters in Table 1, we find $\theta_f \approx 0.5326$, and the importance of this cutoff is reflected in Fig 5.

Particularly, as illustrated in Fig 5A and 5B, when the public response is above $\theta_{\text{critical}}$, infections die out over time with no second wave, and the decay rate positively correlates with the value of $\theta(t)$. Conversely, when there is no adherence to public policy, i.e. $\theta(t > t_n) = 0$, the infections dramatically increase, leading to a large second wave of infections, see Fig 5C. Similarly, we see that when $\theta(t > t_n)$ decays exponentially over time, as in Fig 5D, infection counts initially decrease, but begin to increase once $\theta(t > t_n)$ has dropped below a new $\theta_{\text{critical}}$ threshold. An interesting case that has been considered by governments is to "pulse" disease control measures at regular intervals between two fixed values. From Fig 5E and 5F, we see that the effect of pulsing depends on the specific details of the pulsing strategy. For example, when pulsing between $\theta = 0$ and $\theta = 0.8$, overall, there is a net increase in the infections over the span of three months. However, pulsing between $\theta = 0.4$ and $\theta = 0.8$ at two week intervals leads to the infection numbers decreasing over time.

## Previous pandemic characteristics can inform future response efforts

We handpicked several future $\theta(t)$ schedules in the previous section. Here, we use the functional form of $\theta(t)$, obtained by fitting to Ontario's epidemiological data for $t \in [t_1, t_n]$, to predict potential functional forms for $\theta(t > t_n)$. To begin, we recognise that Fig 3 illustrates $\theta(t)$ exhibits oscillatory behaviour for $t > \tau_2$. Hence, we first consider the case of maintaining this oscillatory behaviour for future times $t > t_n$. Specifically, we assume that $\theta(t) = \theta(t - T)\ \forall\ t > t_n$ where the period $T = t_n - \tau_2$, as illustrated in Fig 6A. This functional form could represent general adherence to public health policies, periodically and temporarily undone by spurious social mixing behaviour. The corresponding infection and fatality trajectories, depicted in Fig 6B, illustrate an oscillatory decay of active cases toward zero in the province over time.

While this type of periodic schedule is intuitive, it is also interesting to consider the case where the oscillations in $\theta(t)$, observed in Fig 3, are instead driven by oscillations in the active

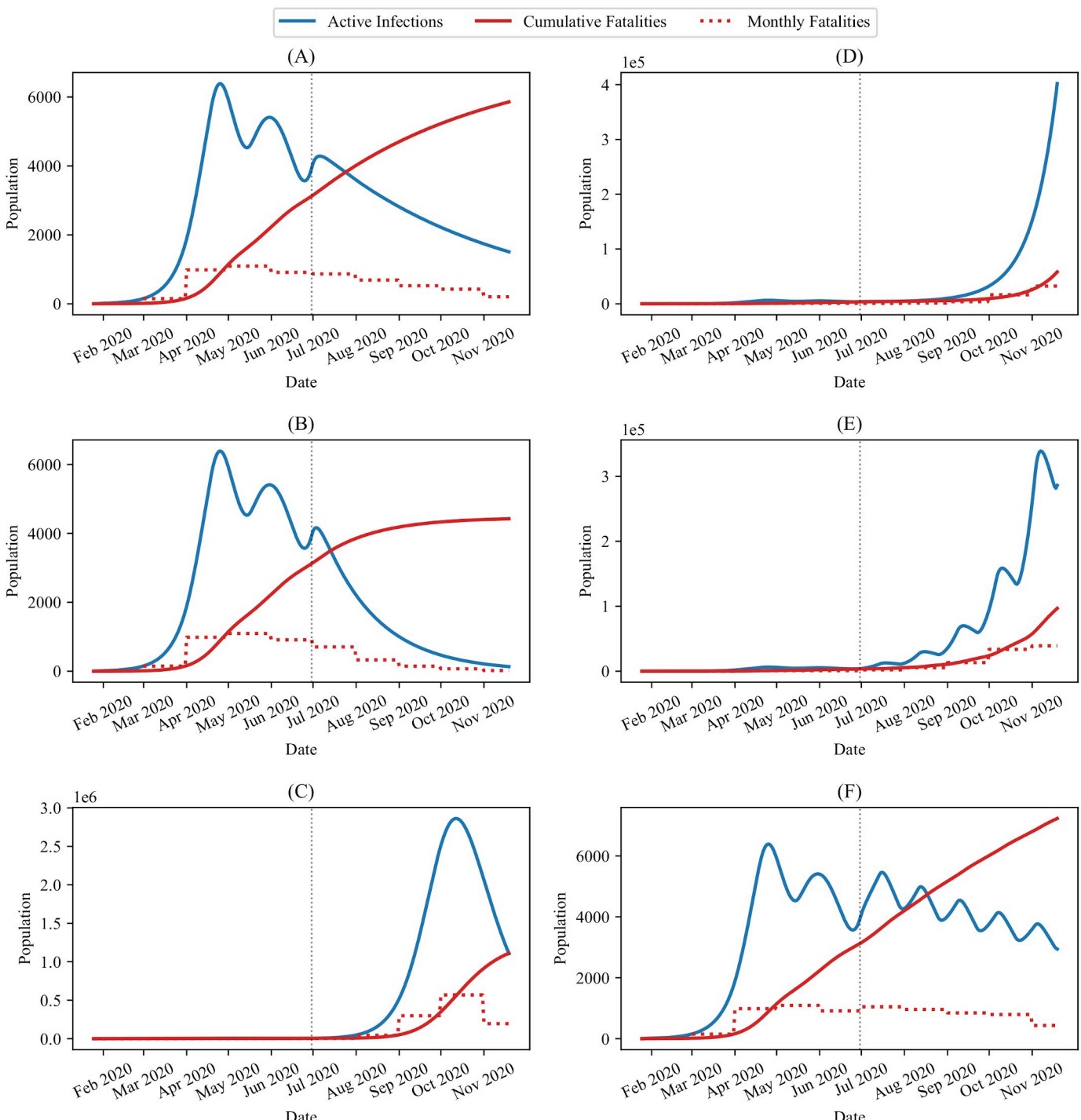

**Fig 5. Potential public response model predictions.** Possible future ($t > t_n$) public response function $\theta(t)$ schedules, where $\theta(t \leq t_n) = 0.6$ and the parameter values and initial conditions are given in Table 1. The blue lines represent the active number of infections, the solid red lines represent the cumulative number of (daily) deaths, and the dashed red lines represent the monthly fatalities. The vertical dashed line is placed at June 30th to differentiate between past and future data. The different future $\theta(t)$ schedules are: (A) Maintain a constant $\theta(t > t_n) = 0.6$, (B) Reduce to $\theta(t > t_n) = 0$, (C) Increase to $\theta(t > t_n) = 0.8$, (D) Exponential decay of $\theta(t > t_n)$ with a half-life of two months, (E) Pulse $\theta(t > t_n)$ between 0 and 0.8 every two weeks, (F) pulse $\theta(t > t_n)$ between 0.4 and 0.8 every two weeks.

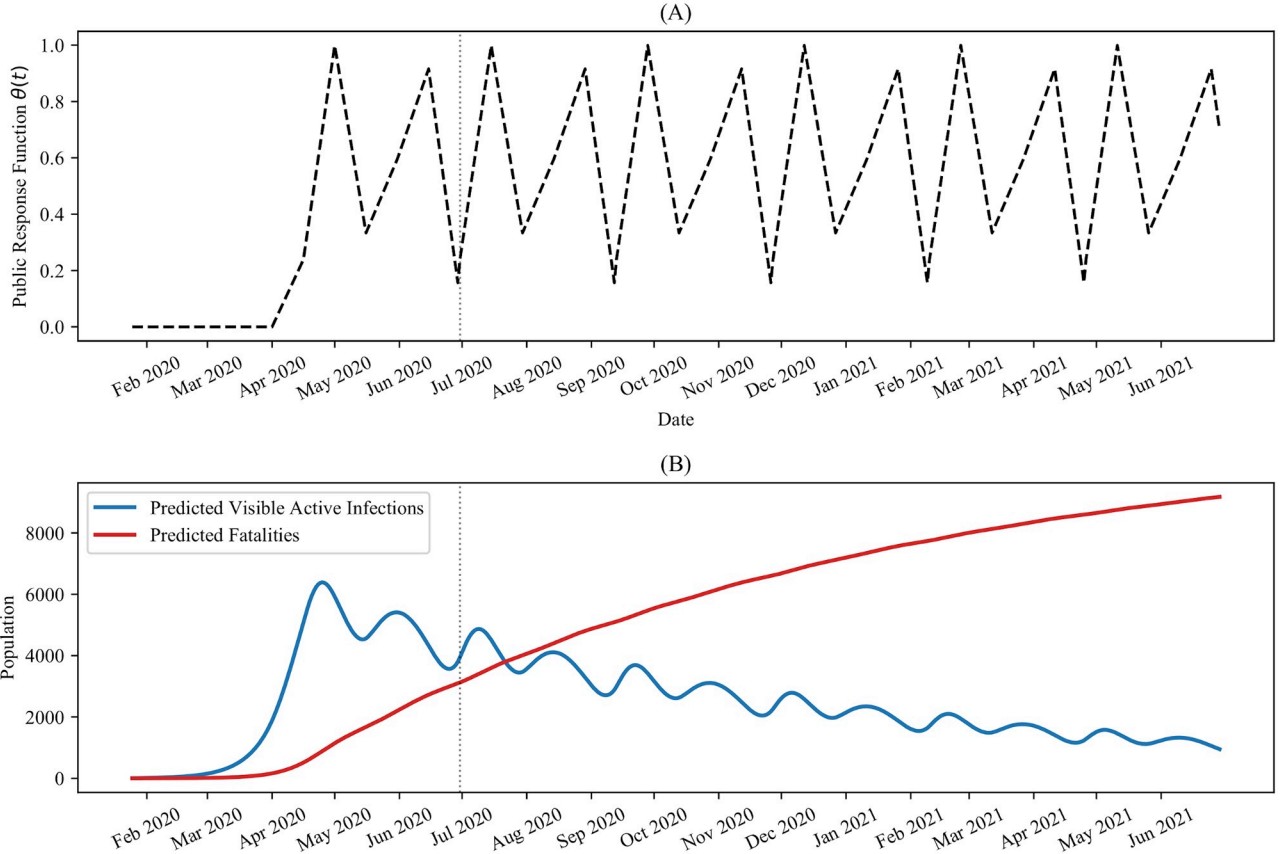

**Fig 6. Predicted pandemic progression for periodic public-response.** Predicted future pandemic progression, obtained by maintaining the periodicity in the functional form of $\theta$, observed for $t \leq t_n$ in Fig 3. (A) The chosen $\theta(t)$ schedule, showing periodic oscillations for $t > t_n$. (B) The corresponding active visible infections and fatalities, predicted by the Distancing-SEIRD model with the parameter values in Table 1. In both plots, the dotted vertical line indicates $\tau_7 = t_n = $ June 30, 2020.

infection numbers. This scenario could correspond to the case where citizens hear news about a dip in active infections, which in turn causes the public to eschew public health transmission reduction policies in favour of social mixing behaviour, thus resulting in an increase in active infections. This new increase in active infections is in turn reported on by the news media resulting in increased vigilance in the populous and stricter adherence to public health policies, eliciting a decrease in active infections, thus causing the cycle to continue. In this hypothetical scenario, we expect the function $\theta(t)$ to effectively depend on the infections, such that $\theta(t) = \theta(I(t))$. Moreover, given the delay between exposure and symptom onset and the delay between symptom onset and confirmed diagnosis, we would anticipate a temporal delay in the dependence of $\theta$ on $I$. That is, $\theta(t) = \theta(I(t-L))$ for some time lag $L$.

To determine the value of the time lag, we calculated the Pearson correlation coefficient between $\theta(t)$ and $I(t-L)$ for several values of $L$, using the epidemiological data for $t < t_n$. The results are presented in Fig 7 and show that the maximum correlation is obtained when $L = 6$, which results in an $R^2$ value of approximately 0.753. We next assumed a linear dependence for $\theta$, with the form $\theta(t) = \min(\max(0, mI(t - 6) + b), 1)$. Using linear regression, we fit this functional form to $\theta(t)$ for $t \leq t_n$, which gave approximately $m = 1.9949 \times 10^{-4}$ and $b = -3.9818 \times 10^{-1}$, with a correlation coefficient of approximately 0.868. The resulting response function is depicted in Fig 8A, and the corresponding predicted infection and fatality curves are illustrated in Fig 8B.

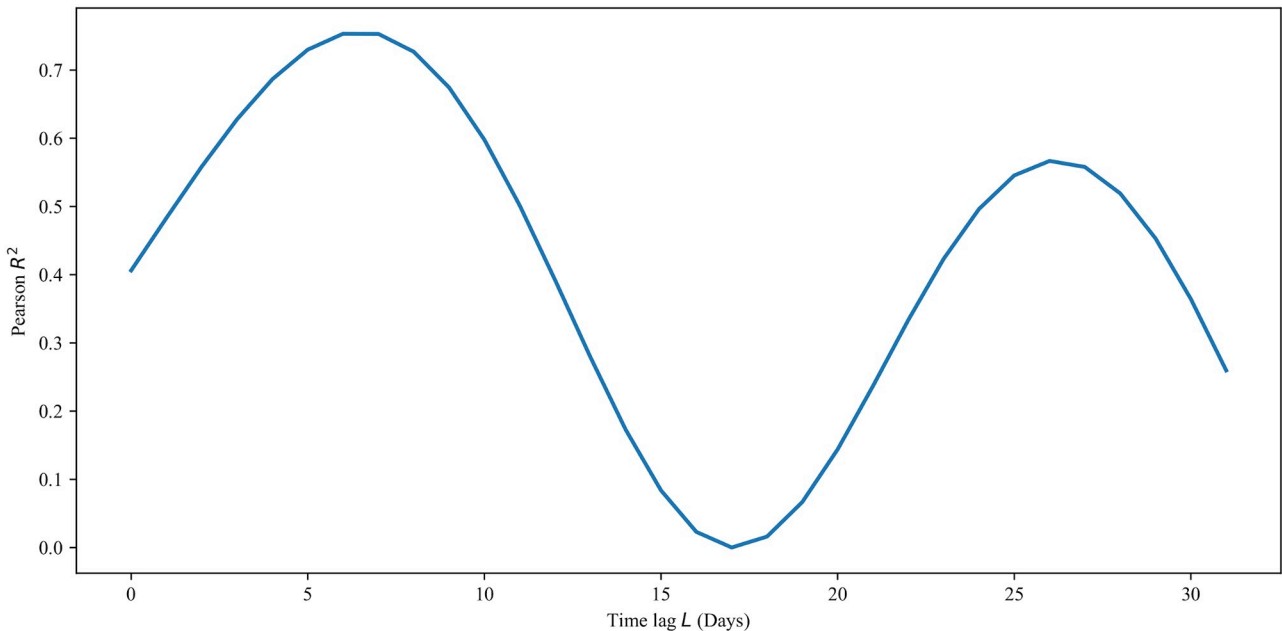

**Fig 7. Pearson $R^2$ between public-response and time-lagged infections as a function of time lag.** The square of the Pearson correlation coefficient, $R^2$, between $\theta(t)$ and $I(t - L)$ for $t \leq t_n$, depicted respectively in Figs 2 and 3, for several values of the time lag $L$.

Importantly, we observe that while this response function is adequate for flattening the active infection curve, it is not enough to fully eradicate the virus in the population on its own.

## Effective reproduction number is sensitive to public response

The basic reproduction number, $R_0$, is the number of expected secondary cases caused by a single infectious individual placed in an otherwise wholly susceptible population. Broadly, the effective reproduction number, $R_t$, represents the expected number of secondary cases seeded by an initial case at time $t$ during the pandemic. The effective reproduction number depends upon properties of the virus and disease itself as well as upon the behaviour of the citizenry in response to public health policy. This number is often argued to be the most important number to track in order to manage a pandemic outbreak. If $R_t < 1$ is sustained, then transmission is slowing and the virus is likely to die out, however if $R_t > 1$ then transmission is increasing within the population. As the pandemic has progressed, local public health officials have enacted various policies to control this number, including social distancing procedures, mandatory face-covering bylaws, and business closures.

Here we used a Bayesian method that was developed by Bettencourt and Riberio [14] to estimate the effective reproduction number $R_t$ from Ontario's epidemiological data. Following this approach, we assumed that new infection cases arise according to a Poisson distribution with mean value equal to the inverse of the serial interval. We further assumed that the distribution of $R_t$ is a Gaussian centred around $R_{t-1}$ with standard deviation $\sigma = 0.1$. Additionally, we made the assumption that the serial interval, for the purpose of calibrating the Poisson distribution, is 4 days [15]. The estimated $R_t$, obtained using this method, is presented in Fig 9 for the SARS-CoV-2 outbreak in Ontario between February 25th, 2020 and June 30th, 2020. While Ontario had its first confirmed case of COVID-19 on January 25th, 2020 the province did not see a sustained increase in case count until late February. The method for estimating $R_t$

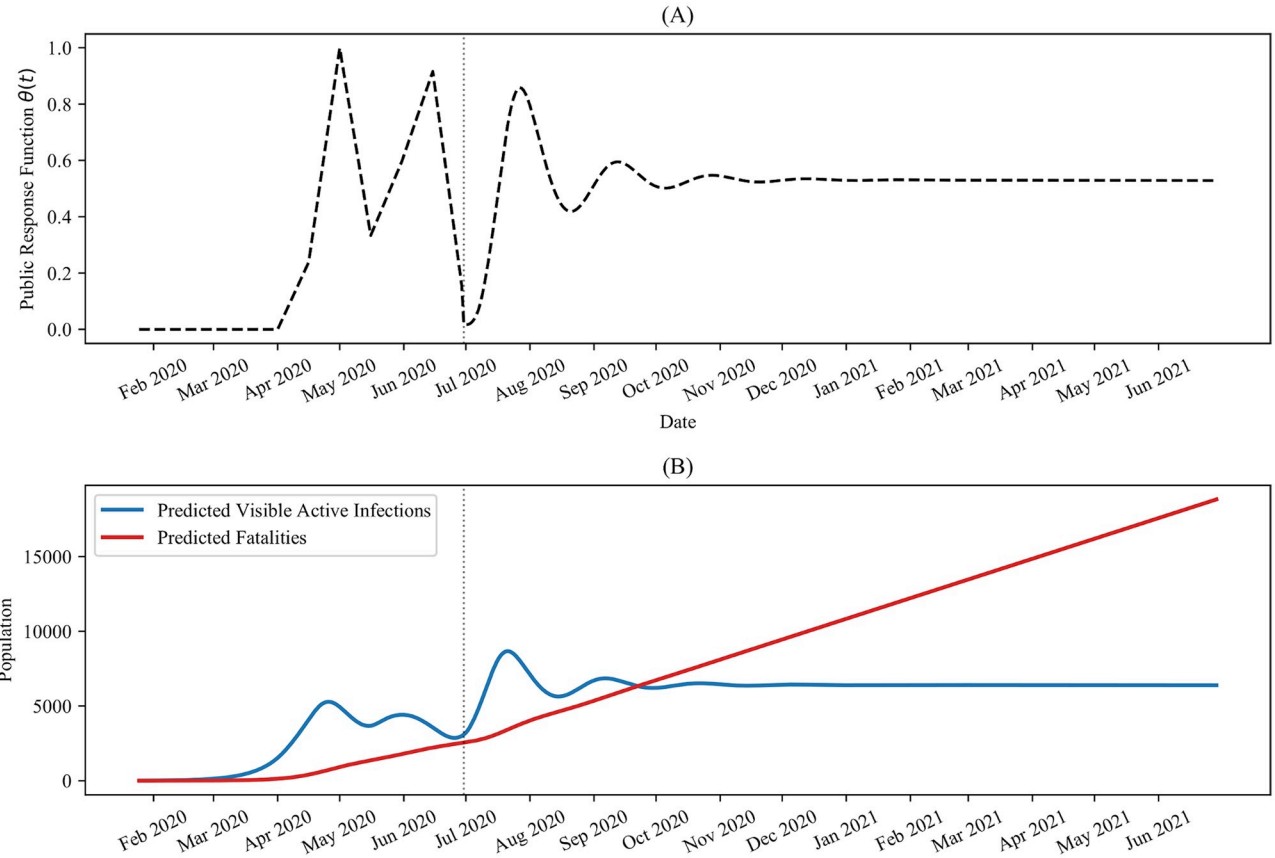

**Fig 8. Predicted pandemic progression for time-lag dependent public-response.** Predicted future pandemic progression, obtained by assuming a linear dependence of $\theta(t)$ on $I(t-6)$. (A) Response function $\theta(t)$ determined by linear regression, (B) The corresponding active visible infections and fatalities, predicted by the Distancing-SEIRD model with the parameter values in Table 1. In both panels, the dotted vertical line indicates $\tau_7 = t_n =$ June 30, 2020.

that we employed requires more than 1 new case a day in a 7-day moving average, as a result we cannot provide estimates for $R_t$ for dates earlier than February 25th, 2020 in Ontario.

We also used the Bayesian method, along with the Distancing-SEIRD model, to predict the trends in $R_t$ for the future $\theta(t)$ functions presented in Fig 5. The results are depicted in Fig 10 and show that the effective reproduction number is quite sensitive to the public response. Specifically, in Fig 10A, which corresponds to maintaining $\theta(t) = 0.6$ for $t > t_n$, $R_t$ ends up slightly below 1 as time progresses, meaning that active infections decay to zero. This result is expected since $\theta_{\text{critical}} = 0.6$. In comparison, in Fig 10B, the public response schedule is $\theta(t) = 0$ for $t > t_n$, corresponding to limited public health containment measures. We see that in this case, $R_t$ spends the entire months of July and August 2020 at around $R_t =$ 1.5. In September, $R_t$ starts decreasing until crossing the $R_t = 1$ line around October 2020. In Fig 5B, there is a rapid increase in active cases before a peak is reached around October 2020. Thus, the qualitative change in $R_t$ around October 2020 in Fig 10B does not correspond to a change in public behaviour, since the public response function remains constant at $\theta(t) = 0$, but rather corresponds to a sufficient (but not total) depletion of the susceptible pool resulting in a decrease in transmission below the $R_t = 1$ bifurcation. Finally, in Fig 10C, corresponding to the public response schedule of $\theta(t) = 0.8$ for $t > t_n$, the behaviour of $R_t$ is similar to the case of $\theta(t) = 0.6$ in Fig 10A. Notably, with higher public response, long-term $R_t$ values

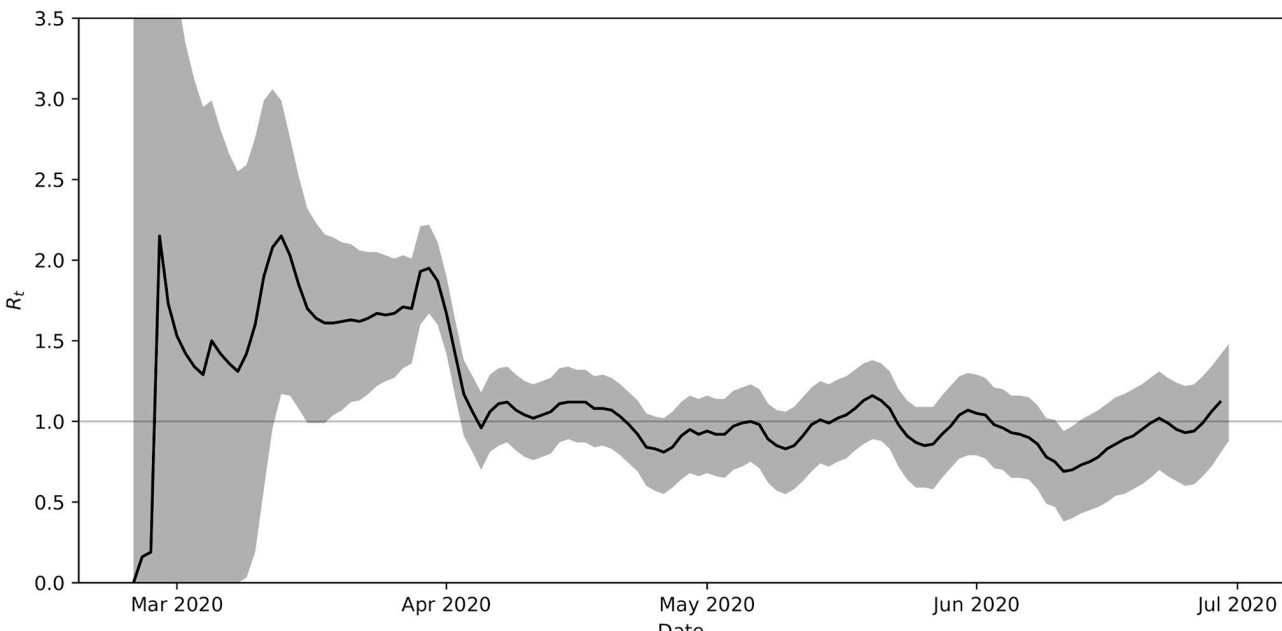

**Fig 9. Effective reproduction number, $R_t$, in Ontario.** Plotted for $t \leq t_n = 157$ days, estimated directly from epidemiological data using the Bayesian method described in Ref. [14]. Grey area represents the 95% confidence interval about the mean values in black.

are smaller and the corresponding active infections decay to zero even quicker, c.f. Fig 5(A) and 5(C).

The cases where $\theta(t)$ was taken to be time-dependent for $t > t_n$ are presented in Fig 10D–10F. We see from Fig 10D that an exponential decay in $\theta(t)$ with a half-life of two months leads to a steady increase in $R(t)$ above $R(t) = 1$ after mid-July 2020. However, the predicted $R(t)$ values in this case remain smaller than those corresponding to $\theta(t) = 0$, Fig 10B, until approximately September 2020. Fig 10E and 10F both correspond to oscillatory schedules for $\theta(t)$ for $t > t_n$, and this periodic behaviour is reflected in the predicted trend in $R_t$. However, while the tails of both $R_t$ plots are oscillatory, only the public response schedule in Fig 10F is capable of reducing the transmission sufficiently to cause a long-term decay in cases, see Fig 5. In contrast, the oscillations in the tail of $R_t$ in Fig 10E result in enough time intervals for which $R_t > 1$ and not enough time intervals for which $R_t < 1$, culminating in an overall long-term increase in cases.

## Conclusion

Mathematical models of pandemic progression can serve as useful tools for policy-makers who are crafting a societal pandemic response. If presented in a widely accessible nature, they can also be used to persuade the general public on best practices for controlling the spread of the disease. One of the main tactics the public can employ to combat the SARS-CoV-2 virus is social distancing: limiting the amount of social interaction in order to lower the effective transmission rate of the virus. Indeed, we saw in this work that the effective reproduction number of the virus is highly sensitive to the public response. Unfortunately, the widespread use of social distancing and other mitigation techniques can cause many economic, social, and personal challenges, leading some to question the value of social distancing in achieving pandemic control. Naturally, questions arise regarding how many people must social distance, and for how long, in order for these tactics to be effective.

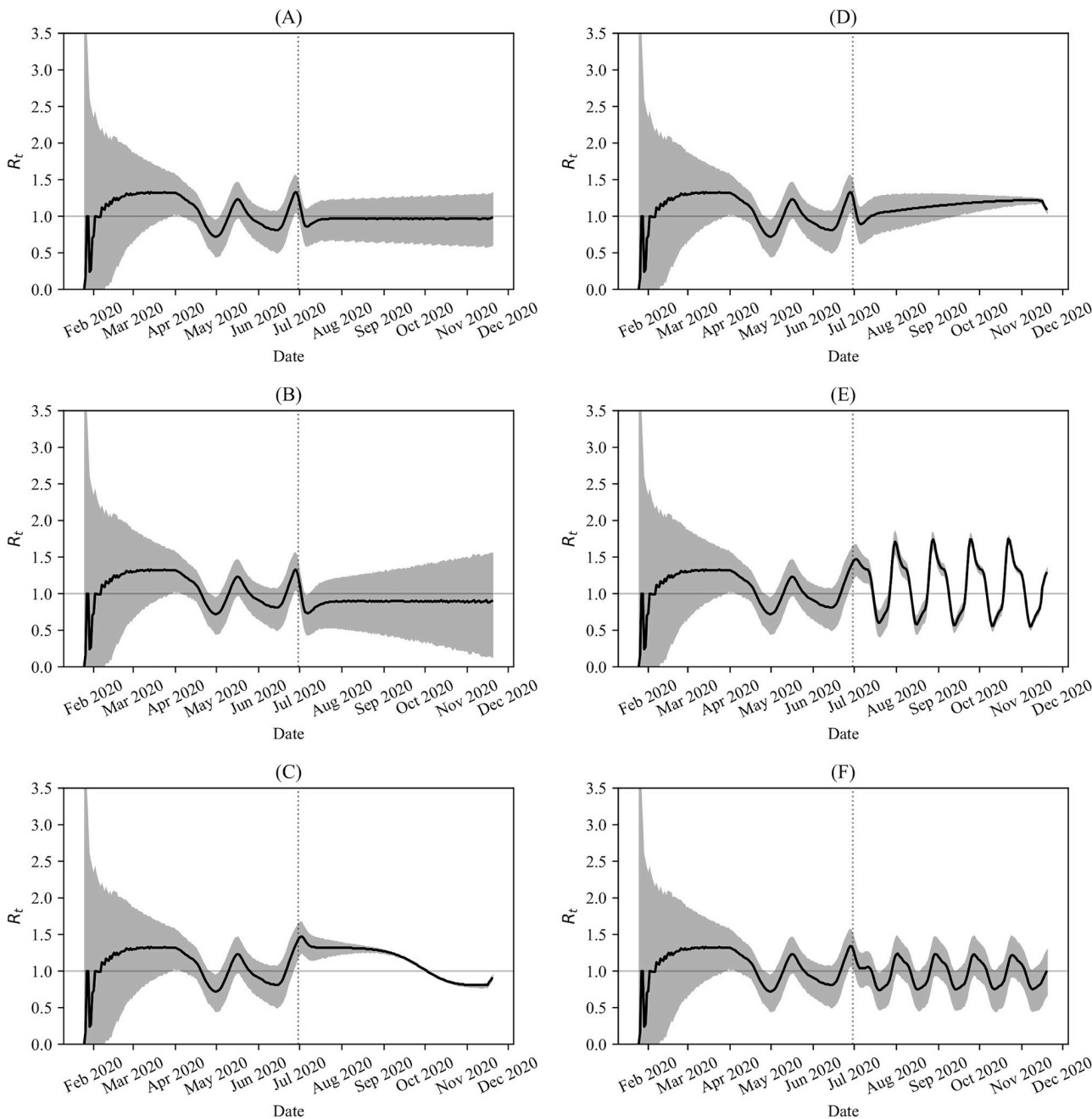

**Fig 10. Predicted $R_t$ evolution for various public-responses.** Plot shows $R_t$ for the future $\theta(t)$ schedules ($t > t_n$) presented in Fig 5, obtained using the Bayesian method described in Ref. [14] along with the Distancing-SEIRD model. Grey area represents the 95% confidence interval about the mean values in black. The dotted vertical line indicates $t_n = 157$ (June 30th), the time value corresponding to the last epidemiological data point used from Ref. [7].

In this work, we presented a mathematical model that is capable of describing the infections, recoveries, and deaths caused by SARS-CoV-2, and can therefore shed light on a number of these questions. The model developed here is a generalized SEIRD model that includes segmentation of the total population into two groups, where one group is adhering to public health measures, while the other is not. The proportion of the population that is following

public health guidelines can be controlled so that time-varying strategies can be examined to make short-term predictions on the spread of the pandemic. This model can be extended to incorporate several different sub-populations, each adhering to a different degree of public health measures. In addition, stochastic effects can be incorporated into the model to account for random events and fluctuations due to individual differences in transmissibility and pre-disposition toward severe disease conditions.

A key result of this work is the development of a method to predict the appearance of a second wave of infections, given the current infection numbers and the disease-specific parameters. Importantly, the appearance of a second wave is found to depend sensitively on the public response. Indeed, we showed in this work that a slight decrease from the critical value of $\theta(t)$ leads to a significant second wave of infections that continues to peak for many months into the future. Importantly, the method for calculating $\theta_{\text{critical}}$ developed here can be used as a benchmark to calculate the approximate level of adherence to public health policies that is necessary in a population to effectively control the viral spread and lead to a prolonged decrease in active infections.

Importantly, the significance of $\theta_{\text{critical}}$ in distinguishing the nature of the future pandemic progression was reflected in our projections of the pandemic. We saw that maintaining super-critical public adherence, i.e. $\theta(t) > \theta_{\text{critical}}$, leads to a decrease in active infections, while reducing public adherence below $\theta_{\text{critical}}$ can lead to an exponential increase in infections and a surge in case fatalities over a period of several months. Increasing $\theta(t)$ well beyond $\theta_{\text{critical}}$ leads quickly to a decline in active cases and a saturation in case fatalities. This means that a near perfect adherence to public health advice could efficiently dampen the pandemic over a period of months. It would be interesting to consider how these results are impacted by spurious injections of infected individuals into the population. This extension of the model could correspond to, for example, infections that are imported into the population due to national or international travel.

In this endeavour, we also saw that time-varying public health measures can be effective at reducing the active infections and flattening the case fatality curve. One particularly effective strategy supported by this work is the pulsing of public response measures, at two-week intervals, between two different values of public adherence. We showed that while this strategy leads to an oscillatory response in the effective reproduction number of SARS-CoV-2 in Ontario, by a suitable combination of the two values, $R_t$ can on average remain below 1. This situation leads to a decay in active case numbers over a period of a few months, while still permitting a degree of socialization. These results are significant because, as we saw in the calculated effective reproduction number that was determined directly from Ontario's epidemiological data, $R_t$ has been hovering around 1 since approximately April 2020. Public adherence to health guidelines will therefore be important, more than ever before, as schools reopen and child care centers move to full capacity, in the midst of the flu season this Fall. If an increase in active infections should ensue, our work provides a method to project the progression of the pandemic under different public responses. These results can be used to devise an efficient plan that can dampen infections, while still permitting the economy to operate at some capacity.

Given the impact of the public response on the spread of the pandemic, we also investigated potential methods for predicting future population behaviour from previous behaviour. We considered the case in which the oscillations that were observed in the epidemiological data were spurred by actions that would remain constant even as active cases decrease. For instance, the oscillations in $\theta(t)$ from Fig 3 could be explained as a natural waxing and waning of responsible pandemic response from the populace. Notably, during the time period over which these oscillations occur, the official public health policy in Ontario was constant. That is, the alternating peaks and valleys could represent general adherence to public health policies

periodically undone by spurious social mixing behaviour as the public's resolve weakens instead of being driven by direct changes to public health policy. This lack of adherence is only temporary however, as the public eventually rebounds back to following public health policies. Interestingly, in this case, maintaining the oscillatory trend was found to cause a slow oscillatory decay in active cases and a saturation in case fatalities in our pandemic projections for Ontario.

We also considered the scenario in which the observed oscillations in the Ontario public response up to June 30, 2020 were driven by oscillations in the infection numbers. As discussed above, this scenario could correspond to the case where people hear news about a dip in active infections, which in turn causes them to engage in social mixing behaviour. This causes an increase in active infections, which is in turn reported on by the news media. The result is a temporary increase in vigilance in the populous and stricter adherence to public health policies, which further elicits a decrease in active infections, thus causing the cycle to continue. We saw that this scenario quickly leads to an increase in active infections, ultimately saturating at a value that is higher than Ontario's peak value prior to June 30, 2020. The pandemic is sustained in this case and the predicted fatalities continue to increase over time. These results reflect the importance of remaining vigilant in following public health guidelines, even when the infection numbers are down and give the false impression that it is safe to engage in social mixing behaviour.

We hope that this work can assist public health officials in crafting policies to lessen the severity of the pandemic, to inform the public on the potential outcomes of the pandemic, and to serve as an impetus for those who are skeptical of following evidence-based public health guidelines. Importantly, we stress that the results presented in this work are estimates that are based upon a simplistic mathematical model that relies on imperfect and incomplete data. Our results should be interpreted qualitatively as an approximation of the possible outcomes of the pandemic given different levels of public adherence to public health measures. A more comprehensive mathematical analysis of the pandemic, which could potentially lead to more accurate quantitative predictions, would require a significantly more complex model and a larger set of data to determine key disease parameters. Indeed, above we have suggested several potential extensions to the model. In addition, the model could be further extended to incorporate the age and sex of people in the population, as well as to account for underlying medical conditions. It would also be interesting to extend the model to include influenza infections, since the flu season typically begins in the Fall and will likely lead to unforeseen challenges during the SARS-CoV-2 pandemic response.

## Appendices

### Appendix 1

**Local sensitivity analysis.**    Here we investigate the sensitivity of the outputs for the Distancing-SEIRD model, Eqs (3) and (4), to perturbations in the nominal parameter set presented in Table 1. To compute the sensitivities, we change the parameter values and the initial values of $E$ and $I$ one-at-a-time by a small amount, $\Delta p$. Here we take $\Delta p$ to be + 1% of the nominal parameter value $p_0$. Since the quantities $E_0$ and $I_0$ represent discrete counts of the population, we take the change $\Delta p$ to be + 1 for these quantities. Then the relative sensitivity of each model population $x = \{S, E, I, R, D\}$ for the parameter or initial condition $p$ is calculated as follows:

$$R_{x,p} = \frac{(x - x_0)/x_0}{(\Delta p)/p_0},\qquad(7)$$

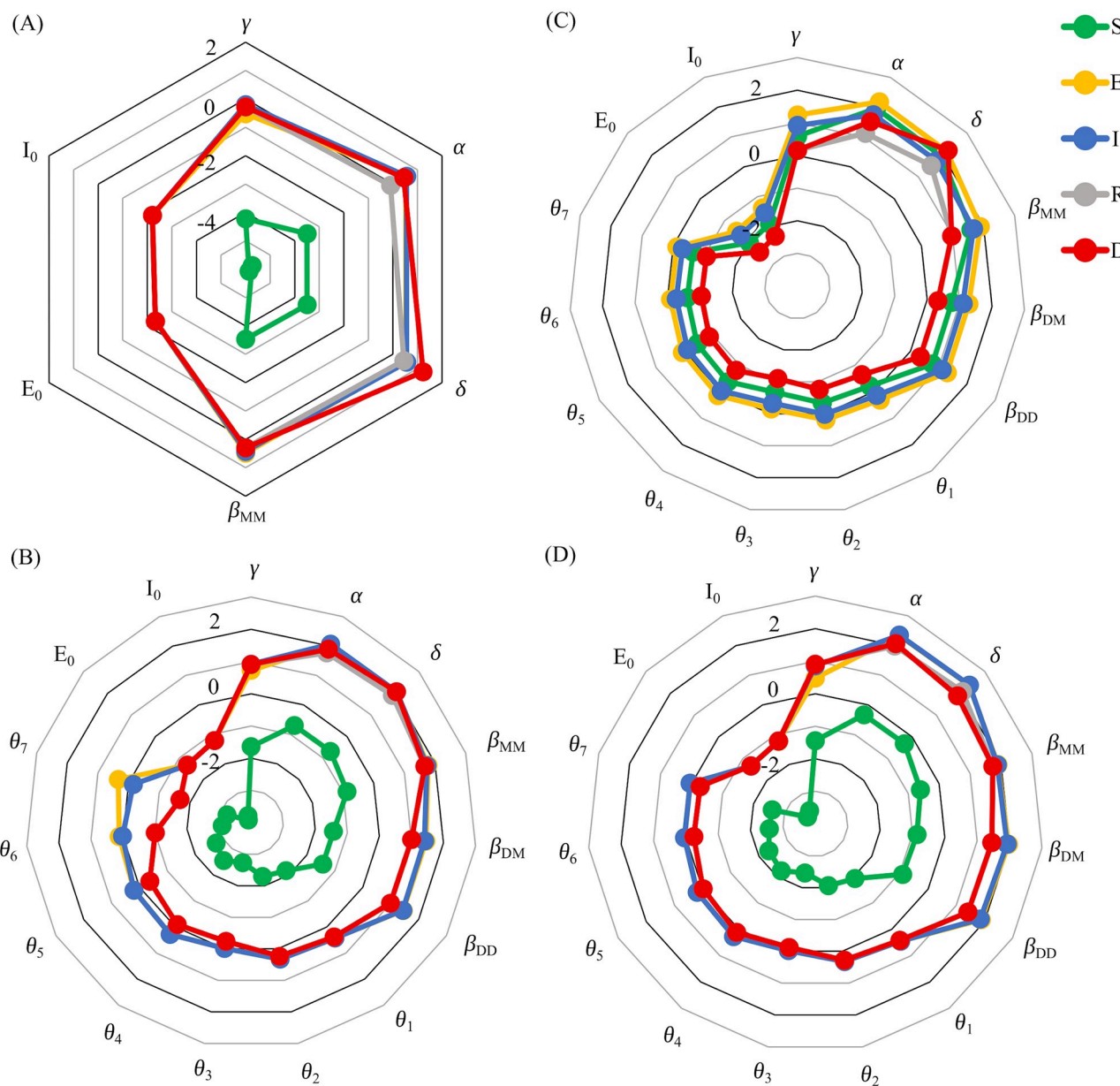

**Fig 11. Relative sensitivities of the outputs of the Distancing-SEIRD model on the fit data.** Perturbations of Eqs (3) and (4), to perturbations in the parameter values and initial conditions. Each parameter was perturbed by + 1% and initial conditions were perturbed by + 1 from their nominal values in Table 1. In all plots, the public response function $\theta(t)$, for $t \leq t_n = 157$ days, is fit from epidemiological data, see Fig 3. (A) Simulation time of 52 days, prior to the start of public health measures. (B) Simulation time of 157 days, capturing the full interval of the epidemiological data. (C) Simulation time of 300 days, where $\theta(t > t_n) = 0$. (D) Simulation time of 300 days, where $\theta(t > t_n) = 0.8$.

where subscripts denote nominal values and $R_{x,p}$ depends on the time through the population compartments.

In Fig 11, we present the calculated relative sensitivities for each of the fitted parameters and non-zero initial conditions in the model. The sensitivities are calculated at the end of the 52 day simulation in Fig 11(A), which corresponds to the date that state of emergency was declared in Ontario. We see here that the susceptible population $S(t)$ is least sensitive of all population compartments to small changes in the parameter values and non-zero initial

conditions. Moreover, the other populations exhibit a fairly strong overlap in their sensitivity values, with the smallest sensitivity to changes in the initial values of $E(t)$ and $I(t)$. The deceased population $D(t)$ is most sensitive to the parameter $\delta$ at this time point. In contrast, we see from Fig 11(B) that further into the pandemic, the sensitivity of $D(t)$ to changes in $\delta$ is lessened and approximately equal for the $E$, $I$, $R$ and $D$ populations. Apart from this parameter, $\alpha$ and $\beta_{MM}$ have the highest sensitivities among all population compartments. Moreover, due to its large size, $S(t)$ is still the least sensitive population compartment by orders of magnitude. Interestingly, the $E$ and $I$ compartments show at least approximately one order of magnitude increase in sensitivity to $\theta_6$ and $\theta_7$ at this time point, compared to the $R$ and $D$ populations.

We consider sensitivity projections, beyond the epidemiological data, in Fig 11(C) and 11(D), where the future response function is taken to have a constant value. We see that in the case where future $\theta(t)$ values are taken to be zero, corresponding to no adherence to public health advice, the susceptible population is thoroughly depleted and becomes more sensitive to perturbations in parameter values and initial conditions than the deceased population. In addition, the exposed population is the most sensitive of all population compartments for all parameters. In comparison, a high adherence to public health policies, such as future $\theta(t) = 0.8$ as in Fig 11(D), leads to a sensitivity pattern that is nearly identical to Fig 11(B), where $\theta(t)$ was fit from the epidemiological data. This pattern is marked by a highly insensitive susceptible population, shown closest to the center of the plot. Unlike Fig 11(B), the $E$, $I$, $R$ and $D$ populations nearly overlap for all parameters, including $\theta_6$ and $\theta_7$.

Importantly, the results in Fig 11 suggest that the relative sensitivities exhibit a qualitatively different trend when the pandemic is winding down compared to when the infection and death counts are high. Particularly, when the pandemic is under control, the susceptible population is nearly insensitive to changes in the parameter values and sits at the center of the radial plot, distinct from the $E$, $I$, $R$ and $D$ populations. However, when the infections are high, the depletion of the susceptibility pool drives up its sensitivity values, resulting in a near overlap in sensitivity trends between all population compartments. This trend is also observed when the future response functions are taken to be time-dependent, as in Fig 12.

Specifically, inspection of the results in Fig 12 reveals that only plots (B) and (D) exhibit the qualitative trend in which the susceptibility pool is robust to changes in the parameter values. Indeed, only the future public response functions chosen in plots (B) and (D) lead to containment of the pandemic, see Fig 5. On the other hand, the other two public response functions lead to an explosion of infections, see Fig 5, and this behaviour is reflected in the trends in the relative sensitivity. Notably, the susceptible population exhibits a much higher sensitivity to all parameter values in plots (A) and (C). We thus see that the qualitative trends in the relative sensitivity of the susceptible population provides a way to determine the state of the pandemic.

## Appendix 2

**Details for the prediction of another wave of infection and an algorithm for determining $\theta_{\text{critical}}$.** If the function $\theta(t)$ is discontinuous, then the functions $S(t)$, $E(t)$, and $I(t)$ remain continuous. However, as can be seen in Eq 3, the functions $S'(t)$ and $E'(t)$ would be discontinuous for discontinuous $\theta(t)$. In the case when $\gamma_D = \gamma_M$, $\alpha_D = \alpha_M$, and $\delta_D = \delta_M$ then by Eq (3) the function $I'(t) = \gamma E - (\alpha + \delta)I$ is entirely independent of $\theta(t)$. Hence, for discontinuous $\theta(t)$ we see that $S(t)$ and $E(t)$ are continuous functions and $I(t)$ is a continuous differentiable function. This is relevant because we may introduce a jump discontinuity in $\theta(t)$ at the point $t_c$ in order to prevent another wave of infection, however $I$ will remain $C^1$ even in the presence of this discontinuity.

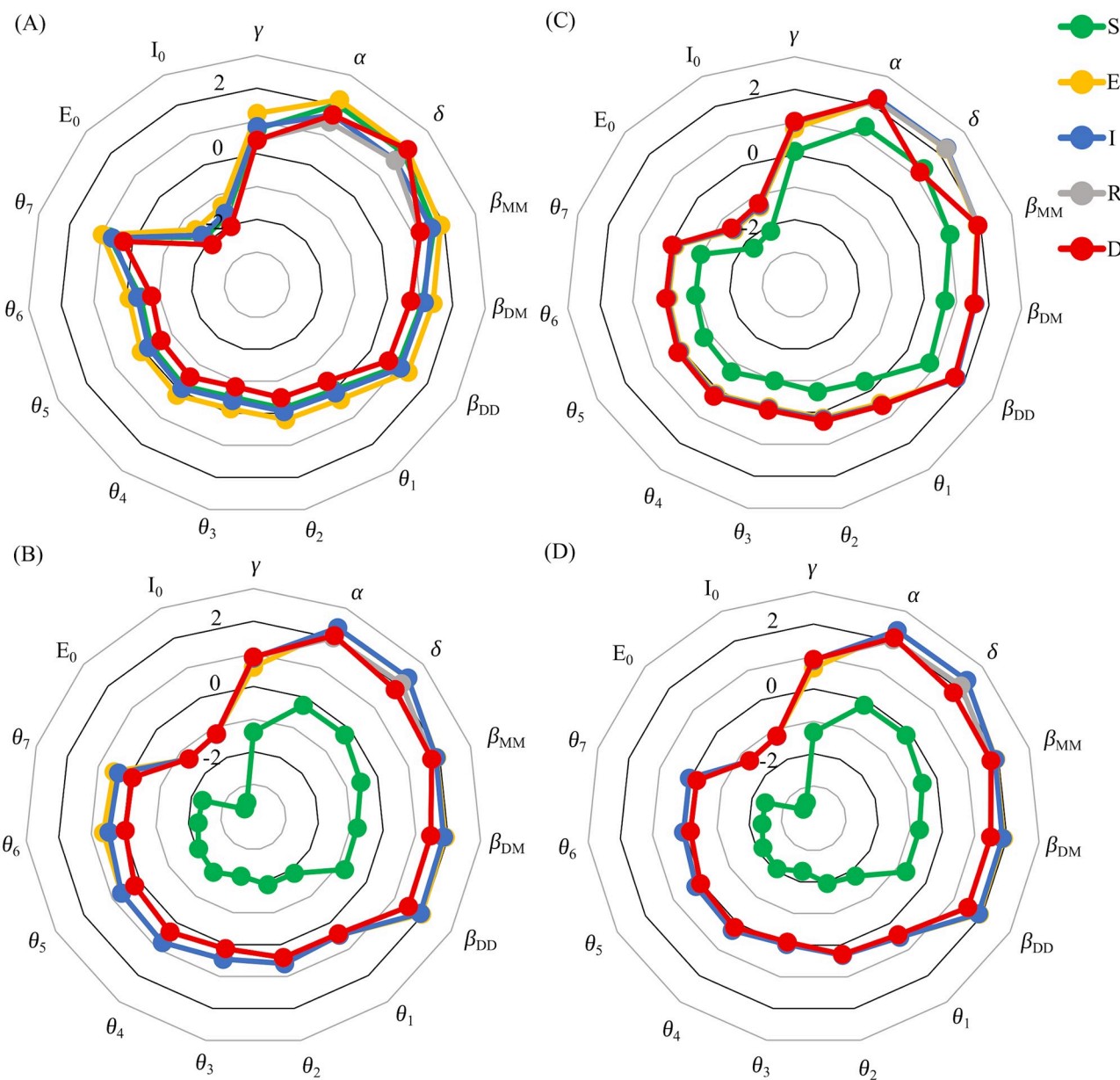

**Fig 12. Projected relative sensitivities of the outputs of the Distancing-SEIRD model.** Sensitivities of Eqs (3) and (4), to perturbations in the parameter values and initial conditions. Each parameter was perturbed by + 1% and non-zero initial conditions were perturbed by + 1 from their nominal values in Table 1. Simulation time in all plots is 300 days. Moreover, in all plots, the public response function $\theta(t)$, for $t \leq t_n$ = 157 days, is fit from epidemiological data, see Fig 3. Local sensitivities were calculated for different time-dependent future public response functions: (A) Exponential decay of $\theta(t > t_n)$ with a half-life of two months. (B) Maintaining periodicity in $\theta(t)$ for $t > t_n$, see Fig 6(A). (C) Pulse $\theta(t > t_n)$ between 0 and 0.8 every two weeks. (D) Pulse $\theta(t > t_n)$ between 0.4 and 0.8 every two weeks.

Suppose that $I'(t_c)>0$, then there exists an open interval $\Omega$ containing $t_c$ such that $I'(t)>0$ for all $t \in \Omega$. As a result, $I(t)>I(t_c)$ for some $t > t_c$. Hence, if $I'(t_c)>0$, then there will always be another wave of infection regardless of the choice of $\theta(t)$ for $t > t_c$. For the remainder of this section we suppose that $\theta(t > t_c) = \theta_f$ for some $\theta_f$ and that $I'(t_c) \leq 0$. We wish to show that if another wave of infection can be prevented for $t > t_c$, then it can be prevented entirely by

choosing $\theta_f \geq \theta_{\text{critical}}$ for some unique value of $\theta_{\text{critical}} \in [0, 1]$ and, conversely, if we choose $\theta < \theta_{\text{critical}}$, then another wave of infection would be inevitable.

Consider $\beta'(\theta) = 2\beta_{DD}\theta + 2(1 - 2\theta)\beta_{DM} + 2\beta_{MM}(\theta - 1)$. Recall that $\beta_{DD} < \beta_{DM} < \beta_{MM}$. Now, $\beta'(0) = 2(\beta_{DM} - \beta_{MM}) < 0$ and $\beta'(1) = 2(\beta_{DD} - \beta_{DM}) < 0$. Hence, since $\beta'(\theta)$ is a linear function of $\theta$, we have established that $\beta'(\theta) < 0$ for $\theta \in [0, 1]$.

As a result, for fixed $t$ we have $\partial_\theta E'(t) = \beta'(\theta)SI < 0$ and $\partial_E I'(t) = \gamma > 0$, hence $\partial_\theta I'(t) < 0$. So if a $\theta_{\text{critical}}$ exists, then it is unique. To see if a $\theta_{\text{critical}}$ can exist for an arbitrary parameter set, we first take $\theta = 1$, solve the system in Eq (3), and use the result with the equation for $I'(t)$ from 3, to see if $I'(t) \leq 0$ for all $t > t_c$. If not, then we have no hope of preventing a second wave as $\partial_\theta I'(t) < 0$. Similarly, we should check if $\theta = 0$ will result in $I'(t) \leq 0$ for all $t > t_c$. If so, then we have the opposite case where any choice of $\theta_f$ will end the infection. In this situation, no optimization on $\theta_f$ is needed. Instead let us suppose that $\theta = 0$ results in an additional wave of infection and $\theta = 1$ results in no additional wave of infection, then, since $\partial_\theta I'(t) < 0$, we are guaranteed the existence of an optimal $\theta_{\text{critical}} \in (0, 1)$ that acts as a separatrix between these two behavioural sets.

To find such a $\theta_{\text{critical}}$ we can use a variant of the bijection method as presented in Algorithm 1. While there are many variant methods that could be used that may be more efficient, the bijection method variant presented in Algorithm 1 is just one example that has the benefit of being simple and is guaranteed to terminate. Moreover, while faster methods can certainly be implemented, Algorithm 1 will approximate $\theta_{\text{critical}}$ within machine epsilon accuracy in 53 iterations (as after 53 iterations, the interval $[\theta_0, \theta_1]$ will have width $2^{-53} \approx 1.11 \times 10^{-16}$ hence the midpoint is at most distance $2^{-54} \approx 5.55 \times 10^{-17}$ from $\theta_{\text{critical}}$).

**Algorithm 1: A Bisection-Like Method for Determining $\theta_{\text{critical}}$.**

```
Input: Parameters t_c, γ, α, δ, β_DD, β_DM, β_MM, and function θ(t) defined
on [0, t_c] for solving the ODE system (3) and a tolerance ϵ.
Output: An approximation θ̂ ∈ [θ_critical − ϵ, θ_critical + ϵ] of the value θ_critical such
that θ > θ_critical avoids a second wave of infection and θ < θ_critical does
not or an error indicating that such a θ_critical does not exist.
1 if I'(t_c) > 0 then
2    return θ_critical does not exist.
3 Solve the ODE system (3) for t > t_c with θ(t > t_c) = 0
4 if I'(t) ≤ 0 for all t > t_c then
5    return θ_critical does not exist.
6 else
7    θ_0 = 0
8 Solve the ODE system (3) for t > t_c with θ(t > t_c) = 1
9 if   I'(t) ≤ 0 for all t > t_c then
10 θ_1 = 1
11 else
12    return θ_critical does not exist.
13 while θ_1 − θ_0 > ϵ do
14    θ_m = (θ_1 + θ_0)/2
15    Solve the ODE system (3) for t > t_c with θ(t > t_c) = θ_m
16    if I'(t) ≤ 0 for t > t_c then
17       θ_1 = θ_m
18    else
19       θ_0 = θ_m
20 return     (θ_0 + θ_1)/2
```

In Fig 13 the example presented is one where $I'(t_c) > 0$. For a very similar case see Fig 4, where $I'(t_c) < 0$. The set up of Fig 13 is similar to that of Fig 4, except $t_c = t_n = 157$ (June 30, 2020) in Fig 13 instead of $t_c = \tau_6 = 142$ (June 15, 2020) as in Fig 4. Note that in Fig 13, no $\theta_f$ strategy presented is capable of entirely avoiding the seeding of another wave of infection.

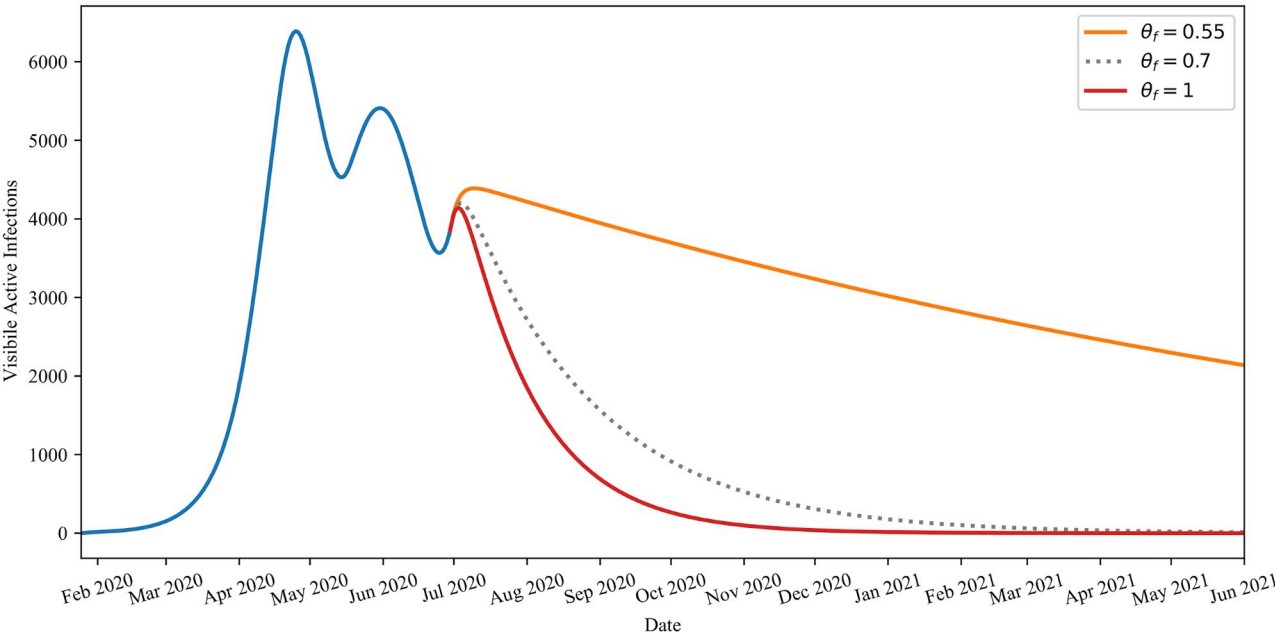

**Fig 13. Demonstration of the conditions on $I'(t_c)$ for inevitable secondary waves.** Behaviour of infections for $\theta_f \in \{0.55, 0.7, 1.0\}$ when $I'(t_c)>0$. This plot demonstrates that no choice of $\theta_f$ is sufficient at preventing an additional wave of infection as even maximum adherence, $\theta_f = 1$, is not enough to prevent $I(t)$ from increasing. Moreover, it illustrates that $\partial_\theta I'(t)<0$.

## Acknowledgments

The authors want to thank the editor and the reviewers for their many helpful comments.

## Author Contributions

**Conceptualization:** Brydon Eastman, Cameron Meaney, Michelle Przedborski.

**Formal analysis:** Brydon Eastman, Michelle Przedborski.

**Investigation:** Brydon Eastman, Cameron Meaney.

**Software:** Brydon Eastman.

**Supervision:** Mohammad Kohandel.

**Writing – original draft:** Brydon Eastman, Michelle Przedborski.

**Writing – review & editing:** Brydon Eastman, Cameron Meaney, Michelle Przedborski, Mohammad Kohandel.

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
