## [Decision Letter · Decision Letter 0]

21 Jan 2021

PONE-D-20-28099

Modeling the impact of public response on the COVID-19 pandemic in Ontario

PLOS ONE

Dear Dr. Eastman,

Thank you very much for submitting your manuscript "Modeling the impact of public response on the COVID-19 pandemic in Ontario" (#PONE-D-20-28099) for review by PLOS ONE. As with all papers submitted to the journal, your manuscript was fully evaluated by academic editor (myself) and by independent peer reviewers. The reviewers appreciated the attention to an important health topic, but they raised substantial concerns about the paper that must be addressed before this manuscript can be accurately assessed for meeting the PLOS ONE criteria. Therefore, if you feel these issues can be adequately addressed, we invite you to submit a revised version of the manuscript that addresses the points raised during the review process. We can’t, of course, promise publication at that time.

We look forward to receiving your revised manuscript.

Kind regards,

Abdallah M. Samy, PhD

Academic Editor

PLOS ONE

**Journal Requirements:**

**Reviewers' comments:**

Reviewer's Responses to Questions

**Comments to the Author**

1. Is the manuscript technically sound, and do the data support the conclusions?

Reviewer #1: Yes

Reviewer #2: Yes

2. Has the statistical analysis been performed appropriately and rigorously? 

Reviewer #1: N/A

Reviewer #2: Yes

3. Have the authors made all data underlying the findings in their manuscript fully available?

Reviewer #1: Yes

Reviewer #2: No

4. Is the manuscript presented in an intelligible fashion and written in standard English?

Reviewer #1: Yes

Reviewer #2: Yes

5. Review Comments to the Author

Reviewer #1: 1. “However, once an individual enters the exposed compartment, they will irreversibly progress to the

infected compartment and subsequently to either the recovered or deceased compartment.”

– Why? They may have been exposed but not infected.

2. “As a result, the infectious period reflected in the epidemiological data is expected to be greater than

14 days, especially since more serious hospitalized cases can take longer than 14 days to resolve in some

individuals. For instance, the World Health Organization reports that the time from onset to recovery is around two weeks for mild cases but is 3-6 weeks for patients with severe or critical disease”

-- What is the average duration used for infection to death?

3. “As powerful as mathematical models are at predicting the spread of infectious diseases, all modelling

is subject to simplifying assumptions to remain tractable. In addition to the assumptions stated above,

the model developed here does not consider how the burden on the healthcare system directly affects the

case fatality rate. In reality, it would be expected that if the healthcare system is overburdened by a large

number of simultaneous infections, the case fatality rate should increase.”

Why did the model disregard hospital occupancy? As stated in the paper itself, it affects case fatality rate which may shorten hospitalization of severely affected/infected patients who may die early in the course of the disease?

4. to “pulse" disease control measures at regular intervals between two _fixed values.

This pulsed control measures at regular interval, also known as circuit breaker, is indeed gaining adherence as a public health that takes into account economic activities. The model, however, does not consider healthcare system utilization rate which forms an important part in preventing mortality.

5. In the Fig 5 graphs, is it possible to graph monthly fatalities rather than cumulative?

6. That is, the alternating peaks and valleys could represent general adherence to public health policies periodically undone by spurious social mixing behaviour as the public's resolve weakens. This lack of adherence is only temporary however, as the public eventually rebounds back to following public health policies. Interestingly, in this case, maintaining the oscillatory trend was found to cause a slow oscillatory decay in active cases and a saturation in case fatalities in our pandemic projections for Ontario.

Is this public rebound to following public health correlated with government-imposed lock down? What social stimulus correlated to the change in behaviour?

7. It would also be interesting to extend the model to include influenza infections, since the u season typically begins in the Fall and will likely lead to unforeseen challenges during the SARS-CoV-2 pandemic response.

If adherence to SARS-CoV-2 public health measures can prevent COVID-19, why should we worry about influenza which is transmitted in similar manner as SARS-CoV-2? In addition, vaccination against influenza is an accepted preventive measure.

Based on several public health data, the incidence of other airbone-transmissible diseases like pneumonia have gone down with the widespread use of mask, disinfection and physical distancing.

Reviewer #2: I recommend publication as this paper is technical sound and have a huge implication for the pandemic.

The modelling is interesting and sophisticated. PLOS one should publish this paper soon during the pandemic.

6. PLOS authors have the option to publish the peer review history of their article (what does this mean?). If published, this will include your full peer review and any attached files.

Reviewer #1: **Yes: **Michael Tee

Reviewer #2: **Yes: **Roger Ho

---

## [Author Response · Author response to Decision Letter 0]

30 Jan 2021

We want to begin by thanking the editor and the reviewers for their careful consideration of our manuscript. We especially want to thank the reviewers for their assessment of our work and reviewer 1 for the many helpful comments. We have addressed the issues raised by reviewer 1 and summarize our response to revisions here. 

In response to the first point, we have added extra language to the manuscript to clarify this distinction. In particular, we assume that anyone who is exposed enough as to be contagious will eventually become infectious. Individuals who are exposed to the virus but do not become contagious then remain in the S (susceptible) category. Implicitly we are assuming that their brief exposure was not enough to acquire antibodies, and so the influence of these individuals is still captured in the model as they still fill the susceptible pool.

In regards to the reviewers second point, we calculated a mean infectious period of approximately 18 days. We have made this calculation more clear in the manuscript.

In response to the third and fourth points, we thank the reviewer for the suggestion and agree that it would be an interesting avenue for future investigation. Including the effect of hospitalization into our model is beyond the scope of the current manuscript, but will be considered in a future work. 

In response to the fifth point, we have made the change in Figure 5. For the sixth point, we have included extra language in the manuscript detailing that while public behaviour changed, official public health policy was constant. This suggests that some other process was behind the shift in behaviour observed.

Finally, for the reviewers seventh point we are grateful for the suggestion. Certainly comorbid infectious diseases that spread via similar mechanisms would be interesting to investigate in general. Such a project would prove to be an enlightening usage of the model we present in this project. However, for the current year influenza rates have fallen significantly (likely as a result of the enhanced public health measures) so the influence of this virus on the spread of SARS-CoV-2 may not be as relevant in our current study.

---

## [Decision Letter · Decision Letter 1]

19 Mar 2021

Modeling the impact of public response on the COVID-19 pandemic in Ontario

PONE-D-20-28099R1

Dear Dr. Eastman,

We’re pleased to inform you that your manuscript, "Modeling the impact of public response on the COVID-19 pandemic in Ontario" (PONE-D-20-28099R1), has been judged scientifically suitable for publication and will be formally accepted for publication once it meets all outstanding technical requirements.

Kind regards,

Abdallah M. Samy, PhD

Academic Editor

PLOS ONE

---

## [Editor Report · Acceptance letter]

24 Mar 2021

PONE-D-20-28099R1 

Modeling the impact of public response on the COVID-19 pandemic in Ontario 

Dear Dr. Eastman:

I'm pleased to inform you that your manuscript has been deemed suitable for publication in PLOS ONE. Congratulations! Your manuscript is now with our production department. 

Kind regards, 

on behalf of

Dr. Abdallah M. Samy 

Academic Editor

PLOS ONE